# Learning to Search for Fast Maximum Common Subgraph Detection

## Abstract

Detecting the Maximum Common Subgraph (MCS) between two input graphs is fundamental for applications in biomedical analysis, malware detection, cloud computing, etc. This is especially important in the task of drug design, where the successful extraction of common substructures in compounds can reduce the number of experiments needed to be conducted by humans. However, MCS computation is NP-hard, and state-of-the-art MCS solvers rely on heuristics in search which in practice cannot find good solution for large graph pairs under a limited search budget. Here we propose GLSEARCH, a Graph Neural Network based model for MCS detection, which learns to search. Our model uses a state-of-the-art branch and bound algorithm as the backbone search algorithm to extract subgraphs by selecting one node pair at a time. In order to make better node selection decision at each step, we replace the node selection heuristics with a novel task-specific Deep Q-Network (DQN), allowing the search process to find larger common subgraphs faster. To enhance the training of DQN, we leverage the search process to provide supervision in a pre-training stage and guide our agent during an imitation learning stage. Therefore, our framework allows search and reinforcement learning to mutually benefit each other. Experiments on synthetic and real-world large graph pairs demonstrate that our model outperforms state-of-the-art MCS solvers and neural graph matching network models.

## 1 Introduction

Due to the flexible and expressive nature of graphs, designing machine learning approaches to solve graph tasks is gaining increasing attention from researchers. Among various graph tasks detecting the largest subgraph that is commonly present in both input graphs, known as Maximum Common Subgraph (MCS) (Bunke & Shearer, 1998) (as shown in Figure 1), is an important yet particularly hard task. MCS naturally encodes the degree of similarity between two graphs, is domain-agnostic, and thus has occurred in many domains such as software analysis (Park et al., 2013), graph database systems (Yan et al., 2005) and cloud computing platforms (Cao et al., 2011). In drug design, the manual testing of the effects of a new drug is known to be a major bottleneck, and the identification of compounds that share common or similar subgraphs which tend to have similar properties can effectively reduce the manual labor (Ehrlich & Rarey, 2011).

MCS detection is NP-hard in its nature and is thus a very challenging task. On one hand, the state-of-the-art exact MCS detection algorithms based on branch and bound run in exponential time in worst cases (Liu et al., 2019). What is worse, they rely on several heuristics on how to explore the search space. For example, MCSP (McCreesh et al., 2017) uses node degree as its heuristic by choosing high-degree nodes to visit first, but in many cases the true MCS contains small-degree nodes. On the other hand, existing machine learning approaches to graph matching such as Wang et al. (2019) and Bai et al. (2020b) either do not address the MCS detection task directly or rely on labeled data requiring the pre-computation of MCS results by running exact solvers.

In this paper, we present GLSEARCH (*G*raph *L*earning to *S*earch), a general framework for MCS detection combining the advantages of search and reinforcement learning. GLSEARCH learns to search by adopting a Deep Q-Network (DQN) (Mnih et al., 2015) to replace the node selection heuristics required by state-of-the-art MCS solvers, leading to faster arrival of the optimal solution for an input graph pair, which is particularly useful when the simpler heuristics fail and graphs are large

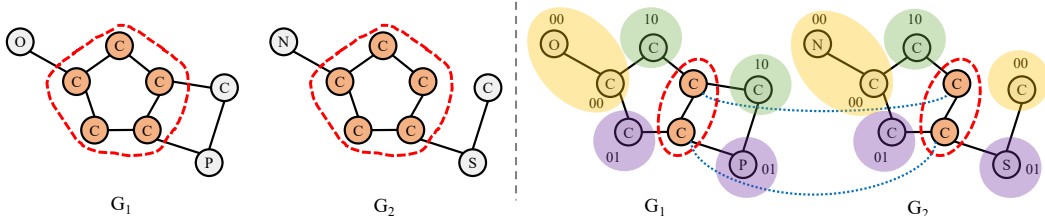

Figure 1: Left: For graph pair $(\mathcal{G}_1, \mathcal{G}_2)$ with node labels, the induced connected MCS is the five-member ring structure highlighted in circle. Right: At this step, there are two nodes currently selected. According to whether each node is connected to the two selected nodes or not, the nodes not in the current solution are split into three bidomains (Section 2.2), denoted as "00", "01", and "10", where "0" indicates not connected to a node in the selected two nodes, and "1" indicates connected. For example, each node in the "10" bidomain is connected to the top "C" node in the subgraph and disconnected to the bottom "C" node in the subgraph.

with a limited search budget. Thanks to the learning capacity of Graph Neural Networks (GNN), our DQN is specially designed for the MCS detection task with a novel reformulation of DQN to better capture the effect of different node selections. Given the large action space incurred by large graph pairs, to enhance the training of DQN, we leverage the search algorithm to not only provide supervised signals in a pre-training stage but also offer guidance during an imitation learning stage.

Experiments on real graph datasets that are significantly larger than exisitng datasets adopted by state-of-the-art MCS solvers demonstrate that GLSEARCH outperforms baseline solvers and machine learning models for graph matching in terms of effectiveness by a large margin. Our contributions can be summarized as follows:

- We address the challenging yet important task of Maximum Common Subgraph detection for general-domain input graph pairs and propose GLSEARCH as the solution.
- The key novelty is the DQN which learns to search. Specifically, it is trained under the reinforcement learning framework to make the best decision at each search step in order to quickly find the best MCS solution during search. The search in turns helps training of DQN in a pre-training stage and an imitation learning stage.
- We conduct extensive experiments on medium-size synthetic graphs and very large real-world graphs to demonstrate the effectiveness of the proposed approach compared against a series of string baselines in MCS detection and graph matching.

## 2 PRELIMINARIES

### 2.1 PROBLEM DEFINITION

We denote a graph as $\mathcal{G} = (\mathcal{V}, \mathcal{E})$ where $\mathcal{V}$ and $\mathcal{E}$ denote the vertex and edge set. An induced subgraph is defined as $\mathcal{G}_s = (\mathcal{V}_s, \mathcal{E}_s)$ where $\mathcal{E}_s$ preserves all the edges between nodes in $\mathcal{V}_s$, i.e. $\forall i, j \in \mathcal{V}_s$, $(i, j) \in \mathcal{E}_s$ if and only if $(i, j) \in \mathcal{E}$. In this paper, we aim at detecting the Maximum Common induced Subgraph (MCS) between an input graph pair, denoted as $\text{MCS}(\mathcal{G}_1, \mathcal{G}_2)$, which is the largest induced subgraph that is contained in both $\mathcal{G}_1$ and $\mathcal{G}_2$. In addition, we require $\text{MCS}(\mathcal{G}_1, \mathcal{G}_2)$ to be a connected subgraph. We allow the nodes of input graphs to be labeled, in which case the labels of nodes in the MCS must match as in Figure 1. Graph isomorphism and subgraph isomorphism can be regarded as two special tasks of MCS: $|\text{MCS}(\mathcal{G}_1, \mathcal{G}_2)| = |\mathcal{V}_1| = |\mathcal{V}_2|$ if $\mathcal{G}_1$ are isomorphic to $\mathcal{G}_2$, $|\text{MCS}(\mathcal{G}_1, \mathcal{G}_2)| = \min(|\mathcal{V}_1|, |\mathcal{V}_2|)$ when $\mathcal{G}_1$ (or $\mathcal{G}_2$) is subgraph isomorphic to $\mathcal{G}_2$ (or $\mathcal{G}_1$).

### 2.2 SEARCH ALGORITHM FOR MCS

Among various algorithms for MCS, we adopt the state-of-the-art search-based algorithm in our framework. The basic version, MCSP, is presented in McCreesh et al. (2017) and the more advanced version, MCSP+RL, is proposed in Liu et al. (2019). The whole search algorithm, outlined in

Algorithm 1[1], is a branch-and-bound algorithm that maintains a best solution found so far throughout the search, which is initialized as empty subgraphs. In each search iteration, denote the current search state as $s_t$ consisting of $\mathcal{G}_1, \mathcal{G}_2$, the current selected subgraphs $\mathcal{G}_{1s} = (\mathcal{V}_{1s}, \mathcal{E}_{1s})$ and $\mathcal{G}_{2s} = (\mathcal{V}_{2s}, \mathcal{E}_{2s})$ as well as their node-node mappings. The algorithm tries to select one node pair, $(v_i, v_j)$, where $v_i$ is from $\mathcal{G}_1$ and $v_j$ is from $\mathcal{G}_2$, as its action, denoted as $a_t$, and either backtracks to the parent search state if the solution is not promising or continues the search otherwise. Various heuristics on node pair selection policy, denoted as "$policy$", are proposed in McSp and McSp+RL. For example, in McSp, nodes of large degrees are selected before small-degree nodes.

There are two major limitations of McSp and McSp+RL: (1) Such heuristics-based node pair selection policy cannot adapt to different graph structures[2]; (2) The search may enter a bad state and get "stuck" without finding a better (larger) solution, $maxSol$, for many iterations.

At each search state, in order to compute the upper bound "$UB_t$" and reduce the action space "$\mathcal{A}_t$", i.e. the candidate node pairs to select from, the concept of "bidomain" is introduced. Bidomains partition the nodes in the remaining subgraphs, i.e. outside $\mathcal{G}_{1s}$ and $\mathcal{G}_{2s}$, into equivalent classes. Among all bidomains of a given state, $\mathcal{D}$, the $k$-th bidomain $D_k$ consists of two sets of nodes, $\langle \mathcal{V}'_{k1}, \mathcal{V}'_{k2} \rangle$ where $\mathcal{V}'_{k1}$ and $\mathcal{V}'_{k2}$ have the same connectivity pattern with respect to the already matched nodes $\mathcal{V}_{1s}$ and $\mathcal{V}_{2s}$. Figure 1 shows an example with three bidomains. Due to the subgraph isomorphism constraint posed by MCS, only nodes in $\mathcal{V}'_{k1}$ can match to $\mathcal{V}'_{k2}$ and vice versa. This also guarantees the extracted subgraphs at each state are isomorphic to each other. Thus, each bidomain can contribute at most $\min(|\mathcal{V}'_{k1}|, |\mathcal{V}'_{k2}|)$ nodes to the future best solution. Therefore, the upper bound can be estimated as $\sum_{D_k \in \mathcal{D}} \min(|\mathcal{V}'_{k1}|, |\mathcal{V}'_{k2}|)$. This upper bound computation is consistently used for all the methods in the paper. The major difference is in the policy for node pair selection.

---

**Algorithm 1** Branch and Bound for MCS

1: **Input:** Input graph pair $\mathcal{G}_1, \mathcal{G}_2$.
2: **Output:** $maxSol$.
3: Initialize stack $\leftarrow$ new Stack().
4: Initialize $maxSol \leftarrow$ empty solution.
5: stack.$push(s_0)$;
6: **while** stack $\neq \emptyset$ **do**
7:    $s_t \leftarrow$ stack.$pop()$;
8:    $curSol \leftarrow s_t.getCurSol()$;
9:    **if** $|curSol| > |maxSol|$ **then**
10:      $maxSol \leftarrow curSol$;
11:    **end if**
12:    $UB_t \leftarrow |curSol| +$ overestimate$(s_t)$;
13:    **if** $UB_t \leq |maxSol|$ **then**
14:      **continue**;
15:    **end if**
16:    $\mathcal{A}_t \leftarrow s_t.$actions;
17:    $a_t \leftarrow policy(s_t, \mathcal{A}_t)$;
18:    $s_t.$actions $\leftarrow s_t.$actions $\setminus \{a_t\}$;
19:    stack.$push(s_t)$;
20:    $s_{t+1} \leftarrow$ env.update$(s_t, \mathcal{A}_t)$;
21:    stack.$push(s_{t+1})$;
22: **end while**

---

## 3   Proposed Method

In this section we formulate the problem of MCS detection as learning an RL agent that iteratively grows the extracted subgraphs by adding new node pairs to the current subgraphs in a graph-structure-aware environment. We first describe the environment setup, then depict our proposed Deep Q-Network (DQN) which provides actions for our agent to grow the subgraphs in a search context. We also describe how to leverage supervised data via pre-training and imitation learning.

### 3.1   Leveraging DQN for Search

Since graph matching for MCS detection must satisfy a hard constraint that the resulting two subgraphs must be isomorphic to each other, instead of learning to match two graphs in one shot, we design an RL agent which explores the input graph pair and sequentially grows the extracted two subgraphs one node pair at a time. The iterative subgraph extraction process can be described by a

---

[1]The original algorithm is recursive. To highlight our novelty, we rewrite into an equivalent iterative version.
[2]McSp+RL claims it uses reinforcement learning (RL) to compute a score for each node pair. However, the scores for each node are the only learnable parameters, whose update relies on a heuristic. This requires the learning of scores for each new testing graph pair. In contrast, we use continuous embeddings to represent graphs which are fed into a DQN to compute a score trained on many graph pairs. Once trained, GLSearch can be applied to any new testing pair without retraining.

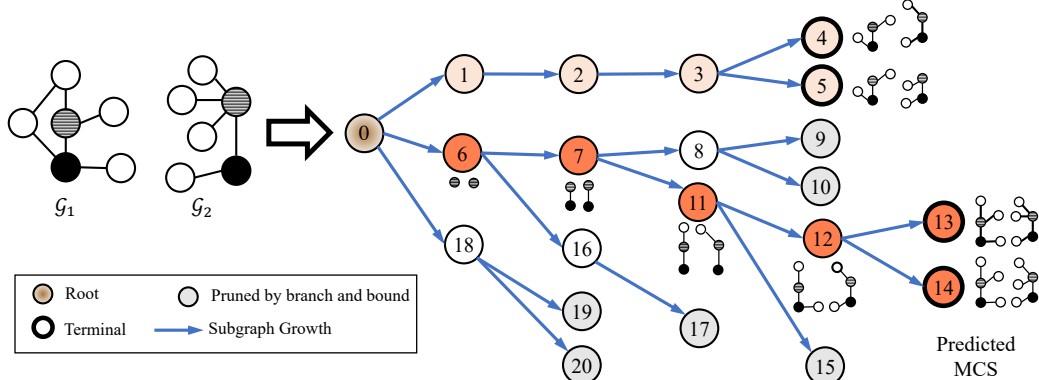

Figure 2: An illustration of the search process for MCS detection. For $(\mathcal{G}_1, \mathcal{G}_2)$, the branch and bound search algorithm (Section 2.2 and Algorithm 1) yields a tree structure where each node represents one state $(s_t)$ with id reflecting the order in which states are visited, and each edge represents an action $(a_t)$ of selecting one more node pair. The search is essentially depth-first with pruning by the upper bound check. Our model learns the node pair selection strategy, i.e. which state to visit first. If state 6 can be visited before state 1, a large solution can be found in less iterations. When the search completes or a pre-defined search iteration budget is used up, the best solution (output subgraphs) will be returned, corresponding to state 13 (and 14).

Markov Decision Process (MDP), where the definitions of state and action are the same as Section 2.2. The difference is that, for MDP, reward needs to be defined too. for MCS, the immediate reward for transitioning from one state to any next state is $r_t = +1$ since one new node pair is selected.

To address the issue that the algorithm may get stuck in a bad state for many iterations without finding a larger solution, we utilize additional information stored in Q-values computed by our learned model. We suppose backtracking to an earlier better state can alleviate such issue in practice, but there lacks a principled measure for MCSP and MCSP+RL to determine which earlier state is better. By design, our node pair selection policy is a learned DQN, so our agent knows not only the quality of immediate actions, but also the values associated with previous states. Therefore, if the best solution found so far does not increase, i.e. we do not enter line 10 of Algorithm 1 for a pre-defined number of iterations, in the next iteration, we find the best state as determined by the Bellman Equation, remove that state, then visit it on line 7. We refer to this improved search methodology as **promise-based search**. More details can be found in the supplementary material.

### 3.2 REPRESENTATION LEARNING FOR DQN

Since the action space can be large for MCS, we leverage the representation learning capacity of continuous representations for DQN design. At state $s_t$, for each action $a_t$, our DQN predicts a $Q(s_t, a_t)$ representing the future reward to go if the action $a_t = (i, j)$ where $i \in \mathcal{V}_1$ and $j \in \mathcal{V}_2$ is selected, intuitively corresponding to the largest number of nodes that will be eventually selected starting from the action edge $(s_t, a_t)$ as in tree in Figure 2.

Based on the above insights, one can design a simple DQN leveraging the representation learning power of Graph Neural Networks (GNN) such as Kipf & Welling (2016) and Velickovic et al. (2018) by passing $\mathcal{G}_1$ and $\mathcal{G}_2$ to a GNN to obtain one embedding per node, $\{\boldsymbol{h}_i | \forall i \in \mathcal{V}_1\}$ and $\{\boldsymbol{h}_j | \forall j \in \mathcal{V}_2\}$. Denote CONCAT as concatenation, READOUT as a readout operation that aggregates node-level embeddings into subgraph embeddings $\boldsymbol{h}_{s1}$ and $\boldsymbol{h}_{s1}$, and whole-graph embeddings $\boldsymbol{h}_{\mathcal{G}_1}$ and $\boldsymbol{h}_{\mathcal{G}_2}$. A state can then be represented as $\boldsymbol{h}_{s_t} = \text{CONCAT}(\boldsymbol{h}_{\mathcal{G}_1}, \boldsymbol{h}_{\mathcal{G}_2}, \boldsymbol{h}_{s1}, \boldsymbol{h}_{s2})$. An action can be represented as $\boldsymbol{h}_{a_t} = \text{CONCAT}(\boldsymbol{h}_i, \boldsymbol{h}_j)$. The Q function can the be designed as:

$$Q(s_t, a_t) = \text{MLP}\big(\text{CONCAT}(\boldsymbol{h}_{s_t}, \boldsymbol{h}_{a_t})\big) = \text{MLP}\big(\text{CONCAT}(\boldsymbol{h}_{\mathcal{G}_1}, \boldsymbol{h}_{\mathcal{G}_2}, \boldsymbol{h}_{s1}, \boldsymbol{h}_{s2}, \boldsymbol{h}_i, \boldsymbol{h}_j)\big). \quad (1)$$

However, there are several flaws to this simple design of Q function:

(A) $\boldsymbol{h}_i$ and $\boldsymbol{h}_j$ generated by typical GNNs encode only local neighborhood information, but $Q(s_t, a_t)$ represents the long-term effect of adding $(i, j)$. What is worse, different node pairs have different embeddings, but their immediate rewards are always $+1$ in MCS.

(B) Swapping the order of $\mathcal{G}_1$ and $\mathcal{G}_2$ should not cause $Q(s_t, a_t)$ to change, but concatenating embeddings from the two graphs causes the DQN to be sensitive to their ordering.

(C) How to effectively leverage the node-node mappings between $\mathcal{G}_{1s}$ and $\mathcal{G}_{2s}$ for predicting $Q(s_t, a_t)$ remains a challenge.

To address these issues, we propose the following improvements over the simple DQN design.

**Factoring out Action**  In order to maximally reflect the effect of adding node pair $(i, j)$ to $\mathcal{G}_{1s}$ and $\mathcal{G}_{2s}$, we first notice that $Q^*(s_t, a_t) = r_t + \gamma V^*(s_{t+1}) = 1 + \gamma V^*(s_{t+1})$ in MCS, where $Q$ and $V$ are the Q and value functions, respectively, and $\gamma$ is the discount factor. Then, in order to compute the effect of $a_t$, we can compute the value associated with $s_{t+1}$ which does not depend on $a_t$ and avoids the use of local $\boldsymbol{h}_i$ and $\boldsymbol{h}_j$.

**Interaction between Input Graphs**  To resolve the graph symmetry issue, we first construct the interaction between the embeddings from two graphs, i.e. INTERACT$(\boldsymbol{h}_{x1}, \boldsymbol{h}_{x2})$, where $\boldsymbol{h}_{x1}$ and $\boldsymbol{h}_{x2}$ represent any embedding from $\mathcal{G}_1$ and $\mathcal{G}_2$ respectively and INTERACT$(\cdot)$ is any commutative function to combine the two embeddings (e.g. summation). This interacted embedding is later concatenated with other useful representations and fed into a final MLP to compute the Q score. We describe our INTERACT$(\cdot)$ operator in Section B.3.

**Bidomain Representations**  Bidomains are derived from node-node mappings and partition the rest of $\mathcal{G}_1$ and $\mathcal{G}_2$, which is a more useful signal for predicting the future reward. In fact, as described in Section 2.2, bidomains have been adopted to in search-based MCS solvers to estimate the upper bound. Here, we require the harder prediction of $Q(s_t, a_t)$ for which we propose to use the representation of bidomains. Denote $\boldsymbol{h}_{D_k}$ as the representation for bidomain $D_k$. Similar to computing the graph-level and subgraph-level embeddings, we compute $\boldsymbol{D}_k$ as

$$\boldsymbol{h}_{D_k} = \text{INTERACT}\big(\text{READOUT}(\{\boldsymbol{h}_i | i \in \mathcal{V}'_{k1}\}), \text{READOUT}(\{\boldsymbol{h}_j | j \in \mathcal{V}'_{k2}\})\big). \tag{2}$$

Since we require the MCS to be connected subgraphs, we differentiate bidomains $\mathcal{D}^{(c)}$ that are connected (adjacent) to $\mathcal{G}_{1s}$ and $\mathcal{G}_{2s}$ from the single bidomain $D_0$ disconnected (unconnected) from $\mathcal{G}_{1s}$ and $\mathcal{G}_{2s}$ (e.g. bidomain "00" in Figure 1). Given all the bidomain embeddings, we compute a single representation for $\mathcal{D}^{(c)}$, $\boldsymbol{h}_{\mathcal{D}c} = \text{READOUT}(\{\boldsymbol{h}_{D_k} | k \in \mathcal{D}^{(c)}\})$. Our final DQN has the form:

$$Q(s_t, a_t) = 1 + \gamma \text{MLP}\Big(\text{CONCAT}\big(\text{INTERACT}(\boldsymbol{h}_{\mathcal{G}_1}, \boldsymbol{h}_{\mathcal{G}_2}), \text{INTERACT}(\boldsymbol{h}_{s1}, \boldsymbol{h}_{s2}), \boldsymbol{h}_{\mathcal{D}c}, \boldsymbol{h}_{\mathcal{D}0}\big)\Big). \tag{3}$$

### 3.3 Leveraging Search for DQN Training

At each state $s_t$, the action space size in the worst case is quadratic to the number of nodes in the remaining subgraphs. Thus, to enhance the training of our DQN, before the standard training of DQN (Mnih et al., 2013), we pre-train DQN and guide its exploration with expert trajectories supplied by the search algorithm.

For the pre-training stage, we first observe the overall mse loss is $(y_t - Q(s_t, a_t))^2$ where $y_t$ the target for iteration $t$ and $Q(s_t, a_t)$ is the predicted $Q(s_t, a_t)$. We then notice that for small training graph pairs, the complete exploration of search space can be performed to obtain the true target for every $(s_t, a_t)$ by finding the longest sequence starting from $s_t$ to a leaf node in the search tree.

For larger graph pairs though, finding the true target becomes too slow. In that case, after pre-training, we enter the imitation learning stage where we let the agent mimic the decision made by the state-of-the-art MCS search algorithm instead of relying on its own predicted $Q(s_t, a_t)$. More details can be in found in the supplementary material.

## 4 Experiments

We evaluate GLSEARCH against two state-of-the-art exact MCS detection algorithms and a series of approximate graph matching methods from various domains. We conduct experiments on a variety of medium-sized synthetic graph datasets and real-world graph datasets, whose details can be found in the supplementary material. Among the different baseline models, we find no consistent trend. This indicates the difficulty of our task, as existing methods can not find a consistent policy that guarantees good performance on datasets from different domains. Our model can substantially outperform the baselines, highlighting the significance of our contributions to learning for search.

Table 1: Results on small and medium graphs. Each synthetic dataset consists of 50 randomly generated pairs labeled as "⟨generation algorithm⟩-⟨number of nodes in each graph⟩". "BA", "ER", and "WS" refer to the Barabási-Albert (BA) (Barabási & Albert, 1999), the Erdős-Rényi (ER) (Gilbert, 1959), and the Watts–Strogatz (WS) (Watts & Strogatz, 1998) algorithms, respectively. NCI109 consists of 100 chemical compound graph pairs whose average graph size is 28.73. We show the ratio of the (average) size of the subgraphs found by each method with respect to the best result on that dataset.

| Method | BA-50 | BA-100 | ER-50 | ER-100 | WS-50 | WS-100 | NCI109 |
|---|---|---|---|---|---|---|---|
| MCSP | 0.913 | 0.892 | 0.842 | 0.896 | 0.905 | 0.856 | 0.948 |
| MCSP+RL | 0.923 | 0.857 | 0.844 | 0.877 | 0.913 | 0.875 | 0.948 |
| GW-QAP | 0.945 | 0.887 | 0.855 | 0.925 | 0.916 | 0.898 | 0.966 |
| I-PCA | 0.899 | 0.863 | 0.848 | 0.923 | 0.879 | 0.852 | 0.951 |
| NEURALMCS | 0.908 | 0.889 | 0.846 | 0.906 | 0.889 | 0.865 | 0.954 |
| GLSEARCH-RAND | 0.995 | 0.987 | 0.920 | 0.978 | 0.967 | 0.931 | 0.989 |
| GLSEARCH | **1.000** | **1.000** | **1.000** | **1.000** | **1.000** | **1.000** | **1.000** |
| BEST SOLUTION SIZE | 19.12 | 34.38 | 26.56 | 37.64 | 29.48 | 55.56 | 10.48 |

## 4.1 BASELINE METHODS

There are two groups of methods: Exact MCS algorithms including MCSP (McCreesh et al., 2017) and MCSP+RL (Liu et al., 2019), learning based graph matching models including GW-QAP (Xu et al., 2019a), I-PCA (Wang et al., 2019), and NEURALMCS (Bai et al., 2020b).

All the methods either originally use or are adapted to use the branch and bound search framework in Section 2.2 with differences in node pair selection policy and training strategies. GW-QAP performs Gromov-Wasserstein discrepancy (Peyré et al., 2016) based optimization *for each graph pair* and outputs a matching matrix $Y$ for all node pairs indicating the likelihood of matching which is treated the same way as our $q$ scores, i.e. at each search iteration we index into $Y$ to select a node pair. I-PCA and NEURALMCS also output a matching matrix but require supervised training, and thus are trained using the same training data graph pairs as our GLSEARCH but with different loss functions and training signals. More details on training and setup of baselines can be found in the supplementary material. During testing, we apply the trained model on all testing graph pairs. For medium-size synthetic testing graph pairs, each method is given a budget of 500 search iterations. For large real-world graph pairs, each method is given a budget of 7500 search iterations. These budgets were chosen based on when the models' performances stabilized. Details about performance using other iteration budgets may be found in the Supplementary Material.

To validate the usefulness of the learned DQN, we compare GLSEARCH, our full model, with a randomly initialized model, GLSEARCH-RAND, which replaces the output of our DQN with a completely random scalar. We show the performance gain of our model through training by substantially outperforming this baseline on all real-world datasets.

## 4.2 PARAMETER SETTINGS

For I-PCA, NEURALMCS and GLSEARCH, we utilize 3 layers of Graph Attention Networks (GAT) (Velickovic et al., 2018) each with 64 dimensions for the embeddings. The initial node embedding is encoded using the local degree profile (Cai & Wang, 2018). We use $\text{ELU}(x) = \alpha(\exp(x)-1)$ for $x \leq 0$ and $x$ for $x > 0$ as our activation function where $\alpha = 1$. We run all experiments with Intel i7-6800K CPU and one Nvidia Titan GPU. For DQN, we use MLP layers to project concatenated embeddings to a scalar. We use SUM followed by an MLP for READOUT and 1DCONV+MAXPOOL followed by an MLP for INTERACT. For training, we set the learning rate to 0.001, the number of training iterations to 10000, and use the Adam optimizer (Kingma & Ba, 2015). The models were implemented with the PyTorch and PyTorch Geometric libraries (Fey & Lenssen, 2019).

## 4.3 RESULTS

The key property of GLSEARCH is its ability to find the best solution in the fewest number of search iterations. As shown in Table 1, our model outperforms baselines in terms of size of extracted subgraphs on all medium-sized synthetic graph datasets and the small chemical compound dataset

Table 2: Results on real-world large graph pairs. Each dataset consists of one large real graph pair ($\mathcal{G}_1$, $\mathcal{G}_2$ may not be isomorphic, but $\mathcal{G}_{1s}$, $\mathcal{G}_{2s}$ are isomorphic guaranteed by search). Below each dataset name, we show its size $\min(|\mathcal{V}_1|, |\mathcal{V}_2|)$ to indicate these pairs are significantly larger than the ones in Table 1. Consistent with Table 1, we show the ratio of the subgraph sizes.

| Method | ROAD 652 | DBEN 1945 | DBZH 1907 | DBPD 1907 | ENRO 3369 | COPR 3518 | CIRC 4275 | HPPI 2152 |
|---|---|---|---|---|---|---|---|---|
| MCSP | 0.374 | 0.815 | 0.797 | 0.722 | 0.694 | 0.684 | 0.498 | 0.864 |
| MCSP+RL | 0.771 | 0.699 | 0.589 | 0.434 | 0.742 | 0.674 | 0.583 | 0.787 |
| GW-QAP | 0.305 | 0.929 | 0.855 | 0.808 | 0.711 | 0.860 | 0.354 | 0.834 |
| I-PCA | 0.267 | 0.551 | 0.589 | 0.607 | 0.650 | 0.707 | 0.203 | 0.762 |
| NEURALMCS | 0.977 | 0.785 | 0.616 | 0.620 | 0.737 | 0.742 | 0.561 | 0.785 |
| GLSEARCH-RAND | 0.641 | 0.762 | 0.658 | 0.639 | 0.814 | 0.755 | 0.603 | 0.814 |
| GLSEARCH | **1.000** | **1.000** | **1.000** | **1.000** | **1.000** | **1.000** | **1.000** | **1.000** |
| BEST SOLUTION SIZE | 131 | 508 | 482 | 521 | 543 | 791 | 3515 | 404 |

NCI109. However, baseline solvers are already quite powerful on these datasets. As it is easy to extract the maximum common subgraph on smaller graph datasets because the total search space grows exponentially with graph size, to truly show the performance advantage of GLSEARCH, we also run experiments on large real-world graph datasets with thousands of nodes.

As shown in Table 10, our model outperforms baselines in terms of the size of the extracted subgraphs on all large real-world datasets. The exact solvers rely on heuristics for node selection, and consistently find smaller subgraphs compared to our results. Figure 3 compares results by MCSP and GLSEARCH. Since MCSP selects nodes with large degrees as its heuristic, the selected nodes tend to be confined in one dense cluster of large degree nodes in $\mathcal{G}_1$. This implies the subgraph in $\mathcal{G}_2$ matched to this dense cluster must also be dense (isomorphism constraint of MCS). In contrast, GLSEARCH is able to find long chains in $\mathcal{G}_1$ which allows easier matching in $\mathcal{G}_2$. In general, there are many cases of large real-world graph pairs where heuristics are not enough to extract large high quality subgraphs. Due to its leveraging both learning and search, GLSEARCH consistently finds subgraphs more than double the size of those found by search based baselines for large real-world graph pairs.

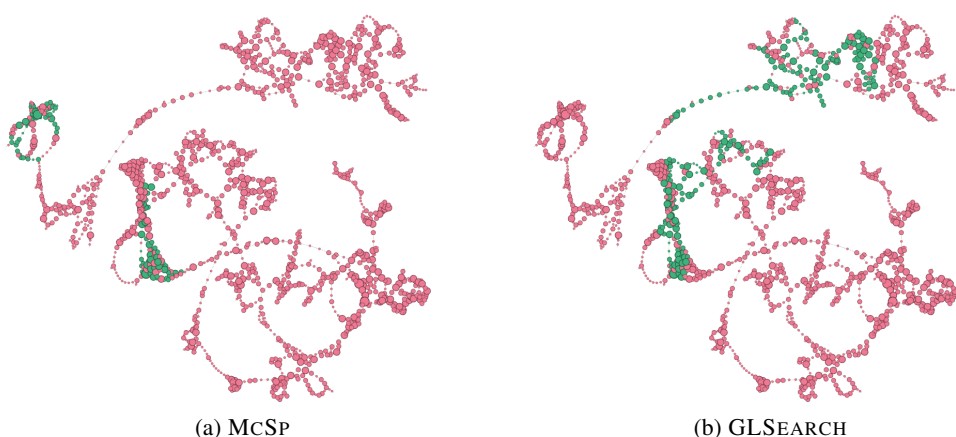

(a) MCSP                    (b) GLSEARCH

Figure 3: Visualization of MCS results on ROAD. Nodes with large degrees have large circles. For each method, we show the two graphs being matched. Selected subgraphs are colored in green.

Compared with learning based graph matching models, GLSEARCH is the only model which learns a reward that is dependent on both state and action, i.e. $Q(s_t, a_t)$. GW-QAP, I-PCA, and NEURALMCS essentially pre-compute the matching scores for all the node pairs in the input graphs, and therefore at each search step, the scores cannot adapt to the particular state, i.e. the matching scores only depend on $\mathcal{G}_1, \mathcal{G}_2$. Notice our state representation includes $\mathcal{G}_1, \mathcal{G}_2$ as well, hence GLSEARCH has more representational power than baselines. Trained under a reinforcement learning framework guided by search, GLSEARCH also performs the best among learning based baselines.

Table 3: Abaltion study on real datasets.

| Method | ROAD | DBEN | DBZH | DBPD | ENRO | COPR | CIRC | HPPI |
|---|---|---|---|---|---|---|---|---|
| GLSEARCH (no $h_{\mathcal{G}}$) | 0.977 | 0.878 | 0.925 | 0.845 | 0.860 | 0.987 | 0.980 | 0.960 |
| GLSEARCH (no $h_s$) | **1.000** | 0.874 | 0.894 | 0.869 | 0.928 | **1.000** | 0.801 | 0.913 |
| GLSEARCH (no $h_{\mathcal{D}c}$) | 0.803 | 0.780 | 0.687 | 0.818 | 0.740 | 0.804 | 0.505 | 0.849 |
| GLSEARCH (no $h_{\mathcal{D}0}$) | 0.576 | 0.856 | 0.782 | 0.768 | 0.823 | 0.932 | 0.323 | 0.938 |
| GLSEARCH (SUM interact) | 0.902 | 0.913 | 0.963 | 0.885 | 0.899 | 0.957 | **1.000** | 0.948 |
| GLSEARCH (unfactored) | 0.447 | 0.807 | 0.712 | 0.582 | 0.816 | 0.816 | 0.512 | 0.861 |
| GLSEARCH (unfactored-i) | 0.500 | 0.789 | 0.741 | 0.772 | 0.748 | 0.825 | 0.902 | 0.864 |
| GLSEARCH | 0.992 | **1.000** | **1.000** | **1.000** | **1.000** | 0.990 | 0.881 | **1.000** |
| BEST SOLUTION SIZE | 132 | 508 | 482 | 521 | 543 | 799 | 3989 | 404 |

## 4.4 ABLATION AND PARAMETER STUDY

To evaluate the effectiveness of different components proposed in our DQN model, we run ablation studies on all real world datasets.

We first measure the importance of each embedding vector fed to our DQN module, as described by Equation 3. We remove each embedding vector (specifically: $h_{\mathcal{G}} = \text{INTERACT}(h_{\mathcal{G}_1}, h_{\mathcal{G}_2})$, $h_s = \text{INTERACT}(h_{s1}, h_{s2})$, $h_{\mathcal{D}c}$, and $h_{\mathcal{D}0}$) individually from the DQN model and retrain the model under the same training settings. Table 3 is consistent with our conclusion that every embedding vector used by GLSEARCH is critical in capturing the search state's representation. Furthermore, we find leveraging bidomain representations is very beneficial to our model.

We next measure the importance of interaction to address the symmetry issue of the MCS calculation, where input graph pairs must be order insensitive. We first test the necessity of using more complex interaction functions, by replacing our 1DCONV+MAXPOOL interaction with simple SUM for interaction (still followed by an MLP). As shown in Table 3, we see that simpler interaction functions may not be powerful enough to encode the interaction between 2 graphs. Particularly, this suggests that interaction is quite important to model performance.

Finally, we measure the importance of factoring out actions from our DQN model. We test this with 2 models. The first utilizes Equation 1 to encode the Q-value, which we refer to as GLSEARCH (unfactored). Since Equation 1 also suffers from the issue of graph symmetry, we adapt this model to use the same interaction function as GLSEARCH to construct 3 order-invariant embeddings $h_{\mathcal{G}} = \text{INTERACT}(h_{\mathcal{G}_1}, h_{\mathcal{G}_2})$, $h_s = \text{INTERACT}(h_{s1}, h_{s2})$, $h_a = \text{INTERACT}(h_i, h_j)$ to concatenate and pass to the final MLP layer in Equation 1. We refer to this model as GLSEARCH (unfactored-i). Our results show that without factoring out the action, our performance is comparable to or worse than MCSP, indicating the significant performance boost introduced by maximally reflecting the effect of adding node pairs.

## 4.5 OVERHEAD OF GLSEARCH

Although in each iteration, GLSEARCH needs to compute a Q score, and is thus more computationally expensive than MCSP and MCSP+RL, the decision GLSEARCH makes is "smarter" so that across many iterations, GLSEARCH can find a larger solution. In order to verify this, for each method, in each search iteration, we collect the best solution (largest subgraph) found so far, and plot the best solution size across search iterations. Since we also measure the running time at each iteration, we also plot the best solution size across running time.

As shown in Figure 4, across the search, initially GLSEARCH finds a smaller solution compared to some baseline methods, but, with more iterations and running time, GLSEARCH quickly outperforms more and more baselines, eventually becoming the best method. Since all the methods follow the same search framework, and the only difference is the node pair selection policy, this experiment verifies that our learned policy can indeed find a larger common subgraph. Given infinite amount of time budget, all the methods would find the true MCS which is the same or larger than the current best solution, but the aim of this section is to show the advantage of the learned policy for search allows the finding of a better solution in fewer iterations and running time. Section E shows the plots on other datasets and provides additional experimental results.

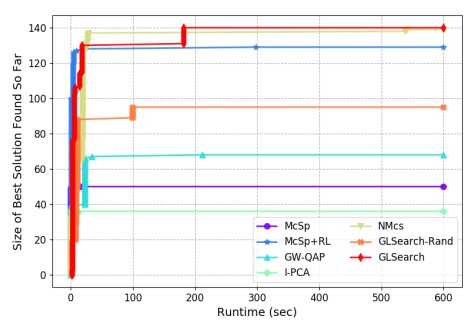 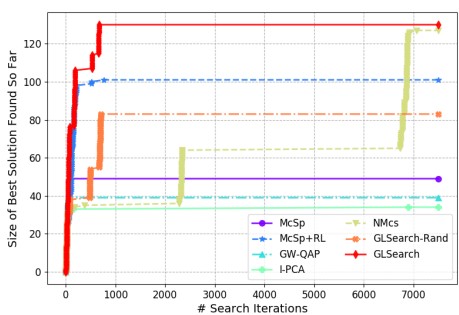

(a) Best solution size across time.    (b) Best solution size across search iterations.

Figure 4: Comparison of the best solution sizes of different methods on ROAD.

## 5 RELATED WORK

MCS detection is HP-hard, with existing methods based on constraint programming (Vismara & Valery, 2008; McCreesh et al., 2016), branch and bound (McCreesh et al., 2017; Liu et al., 2019), mathematical programming (Bahiense et al., 2012), conversion to maximum clique detection (Levi, 1973; McCreesh et al., 2016), etc. Closed related to MCS detection is Graph Edit Distance (GED) computation (Bunke, 1983), which in the most general form refers to finding a series of edit operations that transform one graph to another and has also been adopted in many task where the matching or similarity between graphs is necessary. There is a growing trend of using machine learning approaches to approximate graph matching and similarity score computation, but these works either do not address MCS detection specifically and must be adapted Zanfir & Sminchisescu (2018); Wang et al. (2019); Yu et al. (2020); Xu et al. (2019b;a); Bai et al. (2019; 2020a); Li et al. (2019); Ling et al. (2020), or rely on a large amount of labeled MCS instances Bai et al. (2020b)

## 6 CONCLUSION

We believe the interaction of search and learning is a promising direction for future research, and take a step towards bridging the gap by tackling the NP-hard challenging task, Maximum Common Subgraph detection. We have proposed a reinforcement learning method which unifies search and deep Q-learning into a single framework. By using the search to train our carefully designed DQN, the DQN provides better node selection policy for search to find large common subgraph solutions faster, which is experimentally verified on real-world large graph pairs. In future, the adaptation of our framework to other NP-hard tasks requiring search can be explored.

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

# A   DATASET DESCRIPTION

This section describes the datasets used for evaluating our model and baselines. Section B.1 describes the dataset we use for training GLSEARCH as well as baseline learning based graph matching models.

We use the following real-world datasets for evaluation:

- NCI109: It is a collection of small-sized chemical compounds (Wale et al., 2008) whose nodes are labeled indicating atom type. We form 100 graph pairs from the dataset whose average graph size (number of nodes) is 28.73.

- ROAD: The graph is a road network of California whose nodes indicate intersections and end-points and edges represent the roads connecting the intersections and endpoints (Leskovec et al., 2009). The graph contains 1965206 nodes, from which we randomly sample a connected subgraph of around 0.05% nodes twice to generate two subgraphs for the graph pair $\mathcal{G}_1 = (\mathcal{V}_1, \mathcal{E}_1)$ and $\mathcal{G}_2 = (\mathcal{V}_2, \mathcal{E}_2)$.

- DBEN, DBZH, and DBPD: It is a dataset originally used in a work on cross-lingual entity alignment (Sun et al., 2017). The dataset contains pairs of DBpedia knowledge graphs in different languages. For DBEN, we use the English knowledge graph and sample 10% nodes twice to generate two graphs for our task. For DBZH, we sample around 10% nodes from the knowledge graph in Chinese. For DBPD, we sample once from the English graph to get $\mathcal{G}_1$ and sample once from the Chinese graph to get $\mathcal{G}_2$. Note that although the nodes have features, we do not use them because our task is more about graph structural matching rather than node semantic meanings, and leave the incorporation of continuous node initial representations as future work.

- ENRO: The graph is an email communication network whose nodes represent email addresses and (undirected) edges represent at least one email sent between the addresses (Klimt & Yang, 2004). From the total 36692 nodes, we sample around 10% nodes to generate the graph pair.

- COPR: An Amazon computer product network whose nodes represent goods and edges represent two goods frequently purchased together (Shchur et al., 2018). The graph contains 703655 nodes from which we sample around 0.5% to get the pair we use.

- CIRC: This is a graph pair where each graph is a circuit diagram whose nodes represent devices/wires and edges represent the connecting relations between devices and wires. In other words, each node is either a device or a wire, and the entire graph is bipartite. The two graphs given are known to be isomorphic and we do not perform any sampling. Nodes have labels about the type of the device/wire. In real world, the successful matching of circuit layout diagrams is an essential process in circuit design verification.

- HPPI: It is a human protein-protein interaction network whose nodes represent proteins and edges represent physical interaction between proteins in a human cell (Agrawal et al., 2018). From the 21557 nodes, we sample around  10% nodes to generate the pair used in experiments.

The details of all the graph pairs can be found in Table 4.

Table 4: Details of real-world graph pairs used in evaluating the performance of baseline methods and GLSEARCH.

| Name | Description | $|\mathcal{V}_1|$ | $|\mathcal{V}_2|$ | $|\mathcal{E}_1|$ | $|\mathcal{E}_2|$ | $\frac{|\mathcal{E}_1|}{|\mathcal{V}_1|}$ | $\frac{|\mathcal{E}_2|}{|\mathcal{V}_2|}$ |
|---|---|---|---|---|---|---|---|
| ROAD | Road Network | 1114 | 652 | 1454 | 822 | 1.305 | 1.261 |
| DBEN | Knowledge Graph | 1945 | 1945 | 6242 | 5851 | 3.209 | 3.008 |
| DBZH | Knowledge Graph | 1907 | 1907 | 4856 | 4948 | 2.546 | 2.595 |
| DBPD | Knowledge Graph | 1945 | 1907 | 6242 | 4856 | 3.209 | 2.546 |
| ENRO | Email Communication Network | 3369 | 3369 | 46399 | 50637 | 13.772 | 15.030 |
| COPR | Product Co-purchasing Network | 3518 | 3518 | 56028 | 40633 | 15.926 | 11.550 |
| CIRC | Circuit Layout Diagram | 4275 | 4275 | 6128 | 6128 | 1.433 | 1.433 |
| HPPI | Protein-Protein Interaction Network | 2152 | 2152 | 54910 | 54132 | 25.516 | 25.154 |

For synthetic datasets, we generate graph pairs using the Barabási-Albert (BA) (Barabási & Albert, 1999) algorithm (edge density set to 5), the Erdős-Rényi (ER) (Gilbert, 1959) algorithm (edge density set to 0.08), and the Watts–Strogatz (WS) (Watts & Strogatz, 1998) algorithm (rewiring probability set to 0.2 and ring density set to 4), respectively.

# B   DETAILS ON DQN AND TRAINING GLSEARCH

## B.1   TRAINING DATA PREPARATION: CURRICULUM LEARNING

Curriculum learning (Bengio et al., 2009) is a strategy for training machine learning models whose core idea is to train a model first using "easy" examples before moving on to using "hard" ones. Our goal is to train a general model for MCS detection task which works well on general testing graph pairs from different domains. Therefore, we employ the idea of curriculum learning in training our GLSEARCH. More specifically, we prepare the training graph pairs in the following way:

- Curriculum 1: The first curriculum consists of the easiest graph pairs that are small: (1) We sample 30 graph pairs from AIDS (Zeng et al., 2009), a chemical compound dataset usually for graph similarity computation (Bai et al., 2019) where each graph has less than or equal to 10 nodes; (2) We sample 30 graph pairs from LINUX (Wang et al., 2012), another dataset commonly used for graph matching consisting of small program dependency graphs generated from Linux kernel; (3) So far we have 60 real-world graph pairs. We then generate 60 graph pairs using popular graph generation algorithms. Specifcally, we generate 20 graph pairs using the BA algorithm, 20 graph pairs using the ER algorithm, and 20 graph pairs using the Watts–Strogatz WS algorithm, respectively. Details of the graphs can be found in Table 5. In summary, the first curriculum contains 120 graph pairs in total.

- Curriculum 2: After the first curriculum, each next curriculum contains graphs that are larger and harder to match than the previous curriculum. For the second curriculum, we sample 30 graph pairs from PTC (Shrivastava & Li, 2014), a collection of chemical compounds, 30 graph paris from IMDB (Yanardag & Vishwanathan, 2015), a collection of ego-networks of movie actors/actresses, and generate 20 graph pairs again using the BA, ER, and WS algorithms but with larger graph sizes.

- Curriculum 3: For the third curriculum, we sample 30 graph pairs from MUTAG (Debnath et al., 1991), a collection of chemical compounds, 30 graph paris from REDDIT (Yanardag & Vishwanathan, 2015), a collection of ego-networks corresponding to online discussion threads, and generate 20 even larger graph pairs using the BA, ER, and WS algorithms.

- Curriculum 4: For the last curriculum, we sample 30 graph pairs from WEB (Riesen & Bunke, 2008), a collection of text document graphs, 30 graph paris from MCSPLAIN-CONNECTED (McCreesh et al., 2017), a collection of synthetic graph pairs adopted by MCSP, and generate 20 graph pairs again using BA, ER, and WS algorithms but with larger graph sizes.

For each curriculum, we train the model for 2500 iterations before moving on to the next, resulting in 10000 training iterations in total.

## B.2   TRAINING TECHNIQUES AND DETAILS

### B.2.1   STAGE 1: PRE-TRAINING

For the first 1250 iterations, we pre-train our DQN with the supervised true target $y_t$ obtained as follows:

- For each graph pair, we run the complete search, i.e. we do not perform any pruning for unpromising states. The entire search space is explored, and the future reward for every action can be found by finding the longest path starting from the action to a terminal state. Since graphs are small in the initial stage, such complete search can be affordable. Using Figure 2 in the main text as an example, for the action that causes state 0 to transition to state 6, the longest path is 0, 6, 7, 11, 12, 13 (or 0, 6, 7, 11, 12, 14).

Table 5: Training graph details. For synthetic graphs, "ed", "p", and "rd" represent edge density, rewiring probability, and ring density, respectively.

| Curriculum | Data Source | # Graph Pairs |
|---|---|---|
| **Curriculum 1** | AIDS | 30 |
| | LINUX | 30 |
| | BA:n=16,ed=5 | 20 |
| | ER:n=14,ed=0.14 | 20 |
| | WS:n=18,p=0.2,rd=2 | 20 |
| **Curriculum 2** | PTC | 30 |
| | IMDB | 30 |
| | BA:n=32,ed=4 | 20 |
| | ER:n=30,ed=0.12 | 20 |
| | WS:n=34,p=0.2,rd=2 | 20 |
| **Curriculum 3** | MUTAG | 30 |
| | REDDIT | 30 |
| | BA:n=48,ed=4 | 20 |
| | ER:n=46,ed=0.1 | 20 |
| | WS:n=50,p=0.2,rd=4 | 20 |
| **Curriculum 4** | WEB | 30 |
| | MCSPLAIN-CONNECTED | 30 |
| | BA:n=62,ed=3 | 20 |
| | ER:n=64,ed=0.08 | 20 |
| | WS:n=66,p=0.2,rd=4 | 20 |

- Given the longest path found for each action, we then compute $y_t = 1+\gamma+\gamma^2+...+\gamma^{(L-1)}$, where $\gamma$ is the discount factor set to 1.0, $L$ is the length of the longest path. In the example above, $y_t = 5$, intuitively meaning that at state 0, for the action that leads to state 6, in future the best solution will have 5 more nodes. In contrast, the action 0 to 1 has $y_t = 4$, meaning the action 0 to 6 is more preferred.

- Given the true target computed for each action, we run the mini-batch gradient descents over the mse loss $(y_t - Q(s_t, a_t))^2$, where the batch size (number of sampled actions) is set to 32.

### B.2.2 STAGE 2: IMITATION LEARNING AND STAGE 3

For stage 2 (2500 iterations) and stage 3 (6250 iterations), we train the DQN using the framework proposed in Mnih et al. (2013). The difference is that in stage 2, instead of allowing the model to use its own predicted $Q(s_t, a_t)$ at each state, we let the model make a decision using the heuristics by the MCSP algorithm, which serves as an expert providing trajectories in stage 2. We aim to outperform MCSP eventually after training using our own predicted $Q(s_t, a_t)$ in stage 3.

Here we describe the procedure of the training process. In each training iteration, we sample a graph pair from the current curriculum for which we run the DQN multiple times until a terminal state is reached to collect all the transitions, i.e. 4-tuples in the form of $(s_t, a_t, r_t, s_{t+1})$ where $r_t$ is 1 and $y_t = 1 + \gamma \max_{a'} Q(s_{t+1}, a')$, and store them into a global experience replay buffer, a queue that maintains the most recent $L$ 4-tuples. In our calculations, $L = 1024$. Afterwards, at the end of the iteration, the agent gets updated by performing the mini-batch gradient descents over the mse loss $(y_t - Q(s_t, a_t))^2$, where the batch size (number of sampled transitions from the replay buffer) is set to 32.

To stabilize our training, we adopt a target network which is a copy of the DQN network and use it for computing $\max_{a'} \gamma Q(s_{t+1}, a')$. This target network is synchronized with the DQN periodically, in every 100 iterations.

Since at the beginning of stage 3, the q approximation may still be unsatisfactory, and random behavior may be better, we adopt the epsilon-greedy method by switching between random policy and Q policy using a probability hyperparameter $\epsilon$. Thus, the decision is made as $\arg\max q$ where $q$ is the our predicted $Q(s_t, a_t)$ for all possible actions $(i, j)$ with $1 - \epsilon$ probability; With $\epsilon$ probability,

the decision is random. This probability is tuned to decay slowly as the agent learns to play the game, eventually stabilizing at a fixed probability. We set the starting epsilon to 0.1 decaying to 0.01.

### B.3 DQN PARAMETER DETAILS

In experiments, we use SUM followed by an MLP for READOUT and 1DCONV+MAXPOOL followed by an MLP for INTERACT. Specifically, the MLP has 2 layers down-projecting the node embeddings from 64 to 32 dimensions. Notice that different types of embeddings require different MLPs, e.g. the MLP used for aggregating and generating graph-level embeddings is different from the MLP used for aggregating and generating subgraph-level embeddings.

For 1DCONV+MAXPOOL, we apply a 1-dimensional convolutional neural network to each one of two embeddings being interacted, followed by performing max pooling across each dimension in the two embeddings before feeding into an MLP to generate the final interacted embedding. Specifically, the 1DCONV contains a filter of size 3 and stride being 1. The MLP afterwards is again a 2-layer MLP projecting the dimension to 32. As shown in the main text, such learnable interaction operator brings performance gain compared to simple summation based interaction.

The final MLP takes four components, $h_{\mathcal{G}} = \text{INTERACT}(h_{\mathcal{G}_1}, h_{\mathcal{G}_2})$, $h_s = \text{INTERACT}(h_{s1}, h_{s2})$, $h_{\mathcal{D}c}$, and $h_{\mathcal{D}0}$, each with dimension 32. It consists of 7 layers down-projecting the 128-dimensional input ($32 \times 4$) to a scalar as the predicted $q$ score. For every MLP used in experiments, all the layers except the last use the $\text{ELU}(x)$ activation function. An exception is the final MLP, whose last layer uses the $\text{ELU}(x) + 1$ as the activation function to ensure positive $q$ output.

A subtle point to notice is the necessity of using either nonlinear readout or nonlinear interaction for generating the bidomain representation. Otherwise, if both operators are a simple summation, the representation for all the connected bidomains ($\mathbf{h}_{\mathcal{D}c}$) is essentially the global summation of all nodes in all the connected bidomains. In other words, the nonlinearity of MLP in the readout operation or the interaction operator allows our model to capture the bidomain partitioning information in $\mathbf{h}_{\mathcal{D}c}$.

## C NOTES ON SEARCH

### C.1 COMPARISON WITH MCSP AND MCSP+RL

The key idea of our model is that under a limited search budget, by exploring the most promising node pairs first, search can reach a larger common subgraph solution faster. In other words, for small graph pairs, all baseline models would obtain the exact MCS result as long as the search algorithm runs to complete, i.e. the stack is eventually empty, meaning no more actions to select and no more states to backtrack to (all states have been visited and fully expanded to all possible next states).

However, for large graph pairs, the task is NP-hard, and the complete search becomes nearly impossible. Exceptions exist though: For example, if the pruning condition based on the upper bound estimation is powerful enough to prune many states, i.e. lines 12-14 in Algorithm 1 in the main text, the search may finish in relatively few iterations. However, we observe that the state-of-the-art solvers, MCSP and MCSP+RL, cannot finish completely for all the graph pairs used in testing. Instead of trying to improve the upper bound estimation to be more exact, in this paper, our goal is to learn a better node pair selection policy, i.e. line 17 in Algorithm 1, to replace the heuristics used by baseline solvers.

Notice that our focus on node pair selection policy instead of upper bound estimation implies that a better selection policy would mean the search can quickly find a larger solution and update its best solution found so far $maxSol$. This not only mean when the search budget is used up, the result returned is larger, but also mean that for subsequent iterations (before the iteration limit is reached), more states would be pruned by checking $UB_t \leq |maxSol|$, thus further helping the search. In summary, in our framework, the upper bound computation strategy remains unchanged, yet the successful node pair selection policy benefits the search in two major ways.

Since we use MCSP in the imitation learning stage of training our DQN, and compare with MCSP and MCSP+RL in the main text, we describe their node pair selection heuristics. For MCSP, when entering a new state, it first selects the node with the largest node degree in $\mathcal{G}_1$, and then enumerates through all the nodes in $\mathcal{G}_2$ in descending order of node degrees. In the original implementation

provided by MCSP, this is achieved by recursive function calls. After all the nodes in $\mathcal{G}_2$ are visited, i.e. the depth-first search of all the node pairs $(i, j)$ finishes where $i$ is the largest-degree node in $\mathcal{G}_1$ and $j$ is every node in $\mathcal{G}_2$, the algorithm selects the second largest-degree node in $\mathcal{G}_1$, and repeats the enumeration of nodes in $\mathcal{G}_2$. After all node pairs are exhausted, the function returns and the algorithm essentially backtracks to the parent state. If the current state is the root node in the search tree, the search is complete and the exact MCS is returned. However, as noted earlier, for large graph pairs it is almost impractical to search exhaustively and a budget on the amount of search conducted has to be applied. Thus, which node pairs to visit first matters a lot for successfully extracting a large solution for large input graphs. However, as seen in Figure 10 and 11, in many cases the true MCS does not contain large-degree nodes, since large-degree nodes tend to form more complicated subgraphs which are harder to match in the other input graph compared to simpler subgraphs like a chain. Thus, by visiting large-degree nodes first, MCSP may not always yield a large solution fast.

In contrast, MCSP+RL maintains a promising score for each node and iteratively updates the scores as search visits more states. The update formula is based on the reduction of upper bound for search, where upper bound in an overestimation of future subgraph size. As search makes progress, the scores are updated in each iteration, and nodes which cause large reduction in upper bound computation get large reward. This has the limitation that for each new graph pair, the scores associated with each node must be re-initialized to 0 and re-learned, since there is no neural network and the only learnable parameters are the scores for each node. At the beginning of search, all scores are initialized to 0, and the search has to break the tie using another heuristic, while once trained, our GLSEARCH can be applied to any new testing pair, and at the beginning of search, the learned parameters in GLSEARCH starts to benefit the search. In other words, the whole design of MCSP+RL can be regarded as a search framework with shallow learning (without neural networks or training via back-propagation). GLSEARCH is the first model to use deep learning for node pair selection.

Another limitation of MCSP+RL is that the scores maintained for nodes reflect the potential ability of a node to reduce the upper bound for future iterations in the current search, which is indirect as the MCS aims to find the largest common subgraph, not the reduction of upper bound. Moreover, the upper bound itself is an overestimation of future subgraph size, which may or may not be close enough to the actual best future subgraph size. In contrast, we aim to predict the $q$ score for actions which directly reflect the best future subgraph size. Overall, the lack of deep learning ability causes MCSP+RL not only to re-estimate the scores for the nodes for each new graph pair, but also to resort to the upper bound heuristic for updating the scores.

To ensure the budget on search iterations is applied consistently for all the models evaluated in the main text, we adapt the original recursive implementation of MCSP in C++ to an iterative implementation in Python so that all the models compare with each other in the same programming language and the same search backbone algorithm. To be specific, we check the iteration count at the beginning of every search iteration and early stop the search if the pre-defined budget is reached.

## C.2 TREE VS SEQUENCE

At this point, having illustrated the differences between GLSEARCH and MCSP and MCSP+RL, it is worth clarifying whether GLSEARCH search yields a tree or sequence in different stages. For training stage 1 and 2, as described in Section B.2, our model is just randomly initialized and not well trained, so the pre-training and imitation learning stages use the policy of MCSP instead of using its own predicted $q$ scores. In stage 1, the complete search is performed to provide maximum amount of supervised signals, i.e. $y_t$, but in stage 2, we start using the RL training framework, i.e. experience replay buffer, target network, etc, so we run the agent multiple times until a terminal state is reached, corresponding to a sequence in a tree, which starts from the root node and ends at a leaf node. For stages 2 and 3, since sequences are generated instead of trees, for each sequence, the upper bound check always passes, because the pruning only happens when backtracking is allowed, i.e. a tree is formed. To see this more clearly, recall the pruning only happens if $UB_t \leq |maxSol|$ in Algorithm 1 in the main text. However, during the sequence generation process, $maxSol$ keeps increasing by one each time a new state is reached. Since $UB_t \leftarrow |curSol| + \text{overestimate}(s_t)$, $UB_t > |curSol|$ for non-terminal $s_t$, and $|curSol| = |maxSol|$, and thus $UB_t > |maxSol|$, and thus the pruning never happens.

At the beginning of stage 2, the sequences we collect are usually not long, since the policy is the same as MCSP in stage 2. However, in stage 3, we start using our predicted $Q(s_t, a_t)$ to get such sequences, and at the end of training, when we apply GLSEARCH to testing pairs during inference, as shown in the main text, we perform better than MCSP and all the other baselines. In inference, we completely rely on our predicted $q$ scores as the policy, and a search tree is yielded, although the tree is not complete since the graph pairs are large and a search budget is reached.

## C.3 TERMINAL CONDITIONS

This section discusses on how a terminal state, i.e. a leaf node in the search tree, is determined. Notice our definition of bidomain (equivalence class) does not include node labels, and in each iteration, we allow the matching between nodes in the same bidomain with the same node label. We consider the node labels as additional pruning on each bidomain, i.e. we further only allow nodes with the same label to match within bidomain when considering actions to be fed into DQN. Suppose $\mathcal{G}_1$ and $\mathcal{G}_2$ are connected graphs. There are two cases:

- Case 1: Nodes are unlabeled (or equivalently, all the nodes have the same label). The terminal condition is that there is no non-empty connected (adjacent) bidomains. For example, there are still some adjacent bidomains, but for each bidomain $\langle \mathcal{V}'_{k1}, \mathcal{V}'_{k2} \rangle$, at least one of $\mathcal{V}'_{k1}$ and $\mathcal{V}'_{k2}$ is empty (containing no nodes), so there is no nodes to match in each bidomain. Examples are states 3 and 6 in Figure 6.
- Case 2: Nodes are labeled. For the terminal condition, there may be still some non-empty connected bidomains, but the node labels do not match causing no more node pairs to select from. For example, one bidomain contains $\mathcal{V}'_{k1}$ with C and N as node labels and $\mathcal{V}'_{k2}$ with H as node label. Then essentially there is no more node pairs left.

## C.4 PROMISE-BASED SEARCH: IMPROVING SEARCH WITH BACKTRACKING

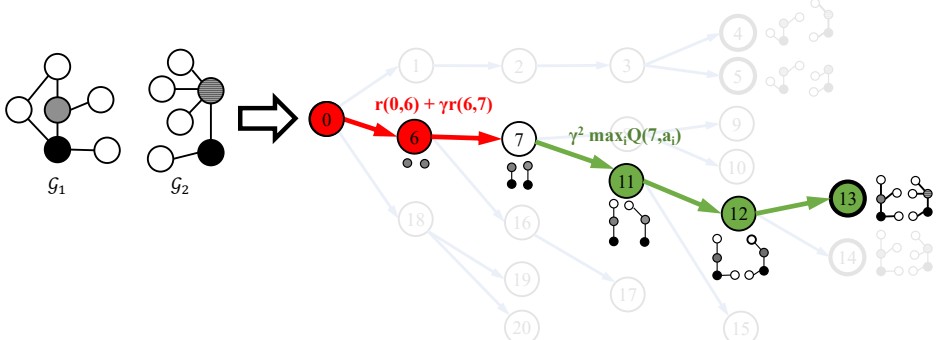

Figure 5: An illustration of how the estimated cumulative sum of rewards is computed for state 7. The expected maximum cumulative sum of rewards obtainable from state 7 is equivalent to the sum of rewards up to state 7 ($r(0, 6) + \gamma r(6, 7)$) and maximum Q-score at state 7 ($\gamma^2 \max_i Q(7, a_i)$). For MCS detection, $r(s_t, s_{t+1}) = 1$ since one new node pair is selected.

Because it computes Q-values, GLSEARCH can leverage more information than baselines when exploring the search space. Specifically, Q-values may be used to compute a predicted maximum cumulative sum of rewards for every search state, $s_t$, in the search tree: $\sum_{j=0}^{t-1} \gamma^j r(s_j, s_{j+1}) + \gamma^t \max_k Q(s_t, a_k)$, where $Q(\cdot, \cdot)$ can be the Q-score predicted by GLSEARCH. This score intuitively encodes maximum subgraph size achievable from every search state, and, for brevity, we will refer call it the promise score. An example of how to compute this score is shown in Figure 5. Because promise scores are comparable across all search states, GLSEARCH can consider the best possible state to visit from any search state, whereas baselines can only consider the best possible state to visit from the current search state. GLSEARCH can also backtrack to more promising search states at any iteration of the search process, whereas past MCS models backtrack only when all future search states from the current state are exhausted.

Unlike MCSP or MCSP+RL, GLSEARCH is optimized to find the largest common subgraph in one try, not to prune the search space. This is because, in practice, even with advanced pruning

techniques, it is not practical to exhaust the entire search space for large graphs. As a consequence, GLSEARCH may fall into local solutions if it strictly follows the MCSP search algorithm (Algorithm 1 of main text). Thus, GLSEARCH improves upon MCSP search by backtracking to an earlier state with the highest predicted maximum cumulative sum of rewards when the current best solution is not improved upon within a fixed number of iterations. In practice, we set this number to 3.

In implementation, whenever a new state, $s_{t+1}$ is reached from state $s_t$ by taking action $a^*$, promise scores for both $s_{t+1}$ and $s_t$ are computed to tell the search algorithm where to backtrack. We associate the promise value of $\sum_{j=0}^{t-1} \gamma^j r(s_j, s_{j+1}) + \gamma^t Q(s_t, a^*)$ to $s_{t+1}$ and update the promise value of $s_t$ with $\sum_{j=0}^{t-1} \gamma^j r(s_j, s_{j+1}) + \gamma^t \max_{a \in A \setminus \{a^*\}} Q(s_t, a)$, because $a^*$ is excluded in future iterations of search from state $s_t$ (line 18 of Algorithm 1 of main text). In addition to pushing states to the stack on line 19 and 21 of Algorithm 1 in the main text, we also push states to an additional priority queue. Thus, whenever the backtrack condition is satisfied, we retrieve states with the highest expected cumulative sum of rewards by popping the priority queue instead of the stack.

## C.5 NOTES ON EQUIVALENT STATES AND MULTIPLE GROUND TRUTHS

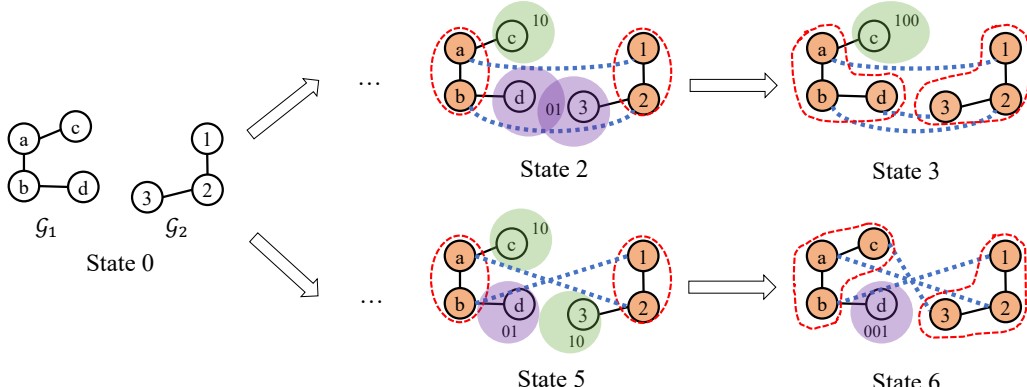

Figure 6: An example illustrating the idea of equivalent states. It is important to note that states 2 3, 5, 6 are different since their node-node mappings are different. However, the solutions derived from both states 3 and 6 have the same subgraph size, 3. In other words, there can be multiple ways to arrive at the same solution size, with different underlying sequential processes to reach the final states.

It is well known that for the MCS detection task, there can be multiple ground truth solutions with the same subgraph size. For example, in Figure 6, both states 3 and 6 correspond to the same subgraph size but the states 3 and 6 are different due to their different node-node mappings. Our model maintains the node-node mappings for each state, and therefore states 3 and 6 would be reached as different states. It is important to note that for large graph pairs, reaching both states 3 and 6 usually does not happen, since the search would first reach state 3 and need many iterations to backtrack to state 0 and further many iterations to reach state 6.

There is an even more subtle point in Figure 6. Suppose the search first matches node a to node 1, denoted as state 1, then matches node b to node 2, leading to state 2. After several iterations, it backtracks to state 0 and chooses to match node b to node 2, denoted as state x, then matches node a to node 1, denoted as state y. Although states 1 and x are different, but states 2 and y are equivalent, since both states 2 and y have the same node-node mapping, i.e. a to 1 and b to 2. Thus, the search maintains an additional set of visited states and at each time a new state is reached, a checking is performed to avoid revisiting the same state twice.

The node-node mapping is is an important component of the definition of state, not only because it differentiates the otherwise equivalent states, but also because different node-node mappings can lead to different final states and future reward (and thus must be considered by the design of DQN). Suppose the bidomain "01" in state 1 contains more than two nodes, i.e. there are many nodes connected to b in $\mathcal{G}_1$ besides node d and many nodes connected to node 2 in $\mathcal{G}_2$ besides node 3. Then state 2 is an intuitively more preferred state compared to state 5, since the matching of node b to node

2 allows more node pairs to be matched to each other in future, thus a larger action space in state 2. The value associated with state 2 thus should be larger than state 5.

## C.6 DEALING WITH LARGE ACTION SPACE

For large graph pairs, the successful detection of MCS not only depend on the design of our critical DQN component as well as its training, but also rely on techniques which prune the large action space at each state.

The bidomain partitioning idea has been outlined in the main text which effectively reduces the action space size by only matching nodes in the same bidomain. However, for extremely large and dense graphs, the bidomains may not split the rest of the graphs enough and the action space may still be too large. For example, consider two fully connected graphs $\mathcal{G}_1$ and $\mathcal{G}_2$, i.e. for every two nodes there is an edge. Then initially, there is no nodes selected, and there is only one bidomain consisting of all the node pairs. What is worse, at any state, there is always only one bidomain consisting of all the node pairs in the remaining subgraphs. Therefore, to reduce the action space further, we only compute the q scores for $N_d$ nodes at most in each state. Specifically, we first sort the candidate bidomains by their size in ascending order, and select the first $N_b$ small bidomains. Next, we sort the nodes in each bidomain by degree in descending order and select the first $N_d/N_b$ nodes with large node degrees. For our experiments, $N_b = 1$ and $N_d = 20$. We find, in practice, these settings do not drastically alter performance. Denote $C_1 = N_d^2$, which is a constant representing the most amount of node pairs the DQN looks at per iteration. Then, in each iteration, the time complexity of GLSEARCH is $\mathcal{O}(C_1)$. Suppose on average, each one of the $C_1$ node pairs requires $C_2$ amount of computations. In practice, for a simple implementation, all the $C_1$ node pairs can be computed sequentially, resulting in $C_1 C_2$ running time. However, various techniques can be applied in practice, e.g. batching the $C_1$ computations, which will further reduce the bounded running time. In implementation, we apply some optimizations to decrease the running time, e.g. computing the node embeddings at the beginning of search and caching them for the entire search during inference. However, we believe further optimization of our implementation is possible and leave for future efforts. In experiments, we set the search budget to 7500 iterations for all baselines to compare and find which method can quickly find a larger solution in terms of iterations.

In future, additional techniques for pruning the action space and speeding up the computation can be explored. For example, instead of bounding the computation to be $N_d$, we may perform a hierarchical graph matching by first running a clustering algorithm and then matching clusters which will be bounded by the number of clusters. Another possible direction is to learn an additional Q function learning which node is more promising instead of which node pair, i.e. $Q(s_t, a_t^{(i)})$ for node $i$ and $Q(s_t, a_t^{(j)})$ for node $j$ in the action $a_t = (i, j)$. We suppose such additional Q function may bring further performance gain.

## C.7 ANALYSIS OF TIME COMPLEXITY

Overall the branch-and-bound search has exponential worst-case time complexity due to the NP-hard nature of exact MCS detection, and our goal is to use additional overhead per search iteration to make "smarter" decision each iteration so that we can find a larger common subgraph faster (in less iterations AND real running time). Per iteration, our model requires the neural network operations to compute a Q score instead of simply using a degree heuristic which is $\mathcal{O}(1)$. Here we analyze the time complexity of these neural operations:

- To compute the node embeddings, the complexity is the same as the GNN model, which in our case is $\mathcal{O}(|\mathcal{V}| + |\mathcal{E}|)$ for GAT (since nodes must aggregate embeddings from neighbors and attention scores must be computed for each edge). Notice the node embeddings are computed by local neighborhood aggregation, and will not be updated in search, and therefore we compute the node embeddings only once at the beginning of search, and can be cached for efficiency.

- At each iteration, to compute a Q score for a state-action pair, we run Equation 3 which requires computing the whole-graph, subgraph, and bidomain embeddings. Overall the time complexity is $\mathcal{O}(|\mathcal{V}| - |\mathcal{V}_s|)$ where $\mathcal{V}_s$ is the number of nodes in the currently matched subgraph. The whole-graph embeddings do not change across search, so they only need to be

computed once at the beginning. The subgraph embeddings can be maintained incrementally, i.e. adding new node embeddings as search grows the subgraph. The bidomain embeddings are computed via a series of READOUT and INTERACT operations (Equation 2): For READOUT: We use summation followed by MLP so the runtime is $\mathcal{O}(|\mathcal{V}| - |\mathcal{V}_s|)$; For INTERACT: We use a 1D CNN followed by MLP which depends on the embedding dimension set to a constant, and does not depend on the number of nodes in the input graphs.

## D  BASELINE DESCRIPTION AND COMPARISON

For all the models used in experiments, we evaluate their performance under the same search framework, i.e. with consistent search iteration counting, upper bound estimation, etc. MCSP and MCSP+RL use heuristics to select node pairs, which is ineffective as shown in the main text and has been described in Section C.1. Therefore, this Section focuses on the comparison with the rest baselines, i.e. GW-QAP (Xu et al., 2019a), I-PCA (Wang et al., 2019), and NEURALMCS (Bai et al., 2020b).

GW-QAP is a state-of-the-art graph matching model for general graph matching. The task is not about MCS specifically, but instead about matching two graphs with its own criterion based on the Gromov-Wasserstein discrepancy (Peyré et al., 2016). Therefore, we suppose the matching matrix $Y$ generated for each graph pair can be used as a guidance for which node pairs should be visited first. In other words, we pre-compute the matching scores for all the node pairs before the search starts, and in iteration, we look up the matching matrix and treat the score as the $q$ score for action selection. I-PCA and NEURALMCS essentially compute a matching matrix too, and it is worth mentioning that all the three methods cannot learn a score based on both the state and the action. They can be regarded as generating the matching scores based on the whole graphs only without being conditioned and dynamically updated on states and actions.

I-PCA is a state-of-the-art image matching model, where each image is turned into a graph with techniques such as Delaunay triangulation (Lee & Schachter, 1980). It utilizes similarity scores and normalization to perform graph matching. We adapt the model to our task by replacing these layers with 3 GAT layers, consistent with GLSEARCH. As the loss is designed for matching image-derived graphs, we alter their loss functions to binary cross entropy loss similar to NEURALMCS which will be detailed below.

NEURALMCS is proposed for MCS detection with similar idea from I-PCA that a matching matrix is generated for each graph pair using GNNs. However, they both require the supervised training signals, i.e. the complete search for training graph pairs must be done to guide the update of I-PCA and NEURALMCS. In contrast, GLSEARCH is trained under the RL framework which does not require the complete search (in stage 2 and 3, only sequences are generated as detailed in Section C.2). This has the benefit of exploring the large action space in a "smarter" way and eventually allows our model to outperform I-PCA and NEURALMCS. In implementation, the complete search is not possible for large training graph pairs, so instead we apply a search budget and use the best solution found so far to guide the training of I-PCA and NEURALMCS.

Regarding the subgraph extraction strategy, for all the baselines, we use the same branch and bound algorithm, which is the state-of-the-art search designed for MCS (McCreesh et al., 2017). However, as mentioned in Section C.4, only our model is equipped with the ability to backtrack in a principled way. The main text shows the performance gain to GLSEARCH brought by the backtracking ability.

## E  MORE RESULTS WITH ANALYSIS

### E.1  BEST SOLUTION SIZES ACROSS TIME

Figure 7 shows that under the budget we set, for the large real-world graph pairs, all the methods reach a "steady state" where the best solution found so far no longer grows. This means the search continues but the search cannot find a larger solution, illustrating the fact that search gets "stuck". In theory, given infinitely many iterations, all the models will eventually find the true MCS, which is the largest, but since the task is NP-hard, the search space can be exponential in the worst case (subject to pruning but in all the testing graph pairs the search has not finished yet), such budget has to be

applied to search. At the end of the 7500 iterations, though, all the models have made "mistakes" (visiting unpromising states) and in order to find even larger common subgraphs, the search needs to backtrack potentially many times to fix those "mistakes" (backtracking to a very early state).

Admittedly, there is always the possibility that for more iterations, some baseline method may find a larger solution. Besides, in each iteration GLSEARCH does take more running time overhead as shown in Table 6. However, the point of our model is to quickly find a larger solution in as few iterations as possible, not to find a large solution given too many iterations. In other words, the goal of GLSEARCH is to be **smart** enough to **quickly** find a large solution instead of purely finding a good solution. Figure 7 shows that GLSEARCH not only finds solutions larger than baselines *when* the 7500 iterations budget is reached but also finds larger solutions faster than baselines *before* the budget is reached.

Table 6: Average running time per iteration (msec).

| Method | ROAD | DBEN | DBZH | DBPD | ENRO | COPR | CIRC | HPPI |
|---|---|---|---|---|---|---|---|---|
| MCSP | 2.040 | 10.724 | 1.415 | 0.974 | 1.722 | 2.891 | 1.776 | 0.498 |
| MCSP+RL | 0.894 | 6.834 | 2.103 | 1.247 | 2.166 | 3.107 | 2.080 | 0.559 |
| GW-QAP | 0.548 | 0.834 | 0.546 | 4.692 | 1.041 | 3.419 | 1.550 | 0.546 |
| I-PCA | 1.152 | 1.797 | 0.967 | 0.897 | 1.739 | 2.725 | 3.792 | 0.636 |
| NEURALMCS | 2.394 | 4.172 | 4.648 | 5.667 | 9.610 | 9.788 | 7.471 | 15.742 |
| GLSEARCH-RAND | 17.392 | 66.418 | 67.342 | 67.946 | 163.005 | 71.972 | 655.447 | 83.488 |
| GLSEARCH | 8.132 | 66.552 | 71.409 | 96.262 | 135.087 | 51.181 | 37.377 | 60.509 |

In addition to reaching a larger solution is less iterations, GLSEARCH also reaches better solutions with respect to runtime, as shown in Figure 8. Notice, GLSEARCH finds the same large subgraph in 10 minutes as in 7500 iterations. Although per iteration, it is slower than MCSP per iteration, GLSEARCH finds a larger solution in usually less than a minute. Moreover, our implementation can be further optimized. Note, we adapted all baselines to run on Python for fair comparison, and made sure the search iteration counting is consistent across all baselines and the results shown in the main text. At this stage, our main goal is to explore the idea of *"learning to search"*, which has been experimentally verified to be a promising direction of research, and leave the efforts of implementation optimization using various techniques as future focus.

## E.2 ADDITIONAL ABLATION STUDY

Table 7 shows that pre-training and imitation learning benefit the performance under four out of the eight datasets. On ENRO and HPPI, without pre-training, our model performs better, which may be attributed to the fact that they are dense graphs (Table 4) while the training graphs used in stage 1 are relatively small and sparse (Section B.1).

Table 7: Contribution of pre-training and imitation learning to the performance of GLSEARCH. "no-sup" denotes the removal of the pre-training stage (The first 3750 iterations: IL; The last 6250 iterations: Normal QDN training); "no IL" denotes the removal of the imitation learning stage (The first 3750 iterations: pre-training; The last 6250 iterations: normal DQN training); "no sup; no IL" indicates the entire training (10000 iterations) is normal DQN training.

| Method | ROAD | DBEN | DBZH | DBPD | ENRO | COPR | CIRC | HPPI |
|---|---|---|---|---|---|---|---|---|
| GLSEARCH (no sup) | 0.557 | 0.957 | 0.946 | 0.904 | **1.000** | 0.999 | **1.000** | **1.000** |
| GLSEARCH (no IL) | **1.000** | 0.933 | 0.965 | 0.887 | 0.357 | 0.875 | 0.666 | 0.632 |
| GLSEARCH (no sup; no IL) | 0.678 | 0.907 | 0.896 | 0.837 | 0.401 | 0.949 | 0.949 | 0.651 |
| GLSEARCH | 0.879 | **1.000** | **1.000** | **1.000** | 0.412 | **1.000** | 0.855 | 0.688 |
| BEST SOLUTION SIZE | 149 | 486 | 465 | 471 | 1318 | 790 | 4112 | 587 |

Table 8 shows that the promised-based search improves the performance under four out of the eight datasets. For the other four datasets, the performance does not change, indicating that the backtracking to an earlier promising state based on the DQN output at least does not hurt the performance. In the cases like ENRO and HPPI, whose average node degrees are large, the promise-based search improves the performance by a large amount, showing the usefulness of the proposed strategy.

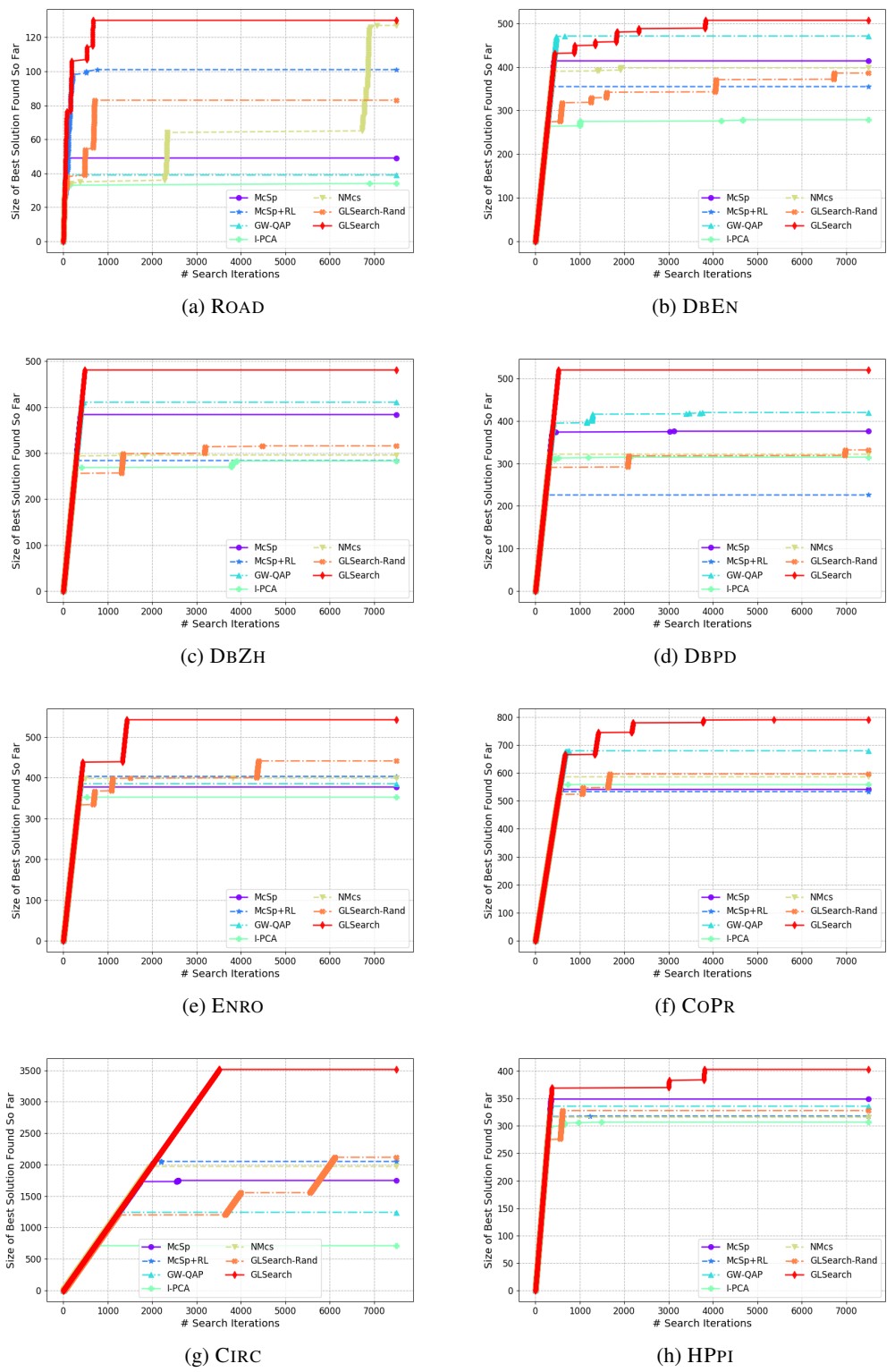

Figure 7: For each method, we maintain the best solution found so far in each iteration during the search process. We plot the size of the largest extracted common subgraphs found so far vs search iteration count for all the methods across all the datasets. The larger the subgraph size, the better ("smarter") the model in terms of quickly finding a large MCS solution under limited budget for large graphs.

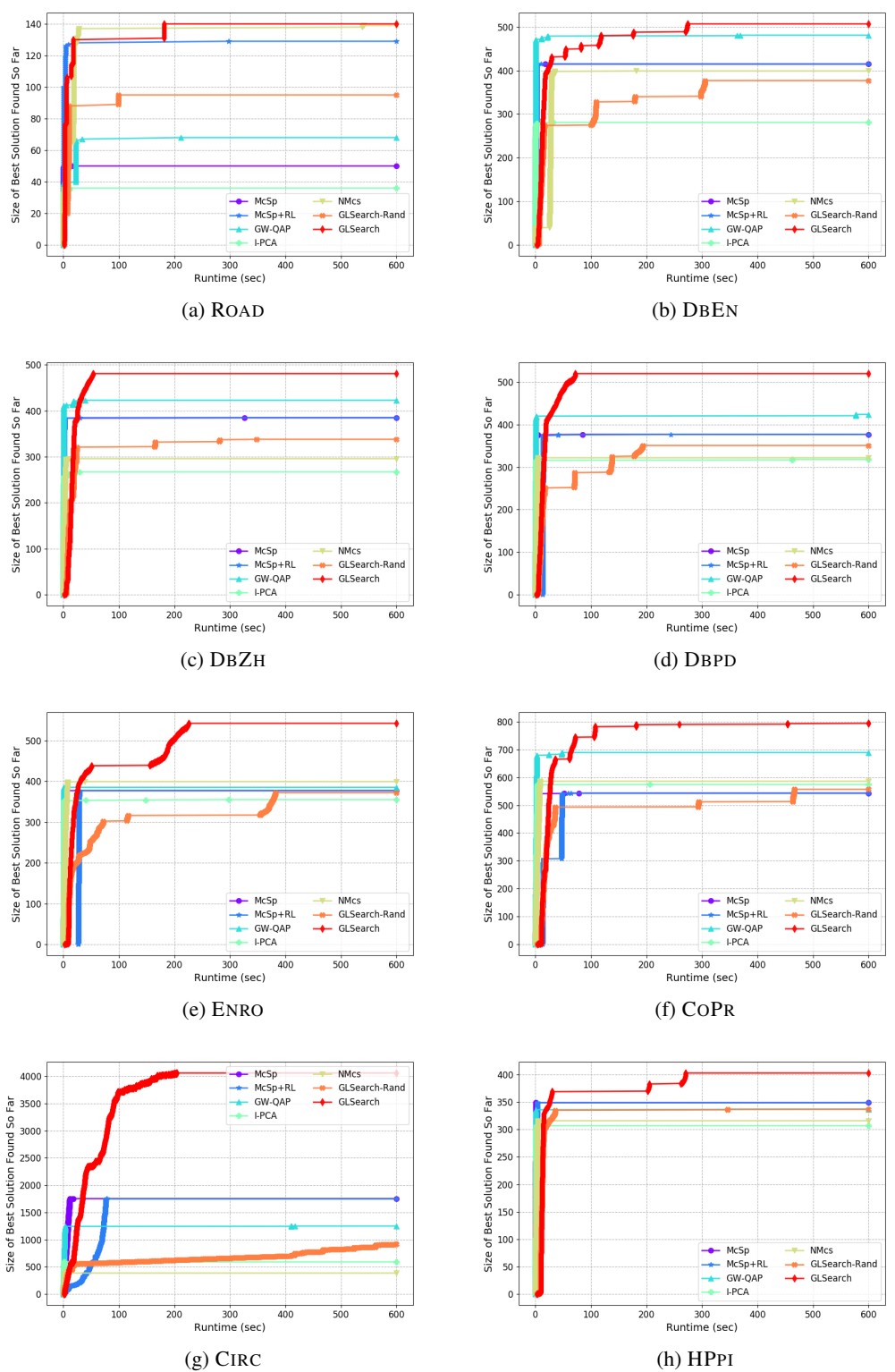

Figure 8: For each method, we maintain the best solution found so far in each iteration during the search process. We plot the size of the largest extracted common subgraphs found so far vs the real running time for all the methods across all the datasets. The larger the subgraph size, the better ("smarter") the model in terms of quickly finding a large MCS solution under limited budget for large graphs.

Table 8: Contribution of promise-based search (Section C.4) to the performance of GLSEARCH. "no promise" denotes that the search does not use the proposed promise-based search, i.e. it does not backtrack to an earlier state if the search makes no progress after a certain amount of iterations, and instead, it continues the regular branch and bound search.

| Method | ROAD | DBEN | DBZH | DBPD | ENRO | COPR | CIRC | HPPI |
|---|---|---|---|---|---|---|---|---|
| GLSEARCH (no promise) | 0.879 | **1.000** | **1.000** | **1.000** | 0.412 | **1.000** | 0.855 | 0.688 |
| GLSEARCH | **1.000** | **1.000** | **1.000** | **1.000** | **1.000** | **1.000** | **1.000** | **1.000** |
| BEST SOLUTION SIZE | 131 | 508 | 482 | 521 | 543 | 791 | 3515 | 404 |

## E.3  SCALABILITY STUDY

GLSEARCH can scale to very large graphs containing tens of thousands of nodes. To verify this, we generate 6 large synthetic graphs (3 graph pairs) using the BA, ER, and WS algorithms, each with 10000 nodes, and ran GLSEARCH against heuristic-based baselines for 7500 iterations. Our model is superior in all instances, as shown in Table 9.

Table 9: Results of GLSEARCH to even larger networks. Each synthetic dataset consists of one graph pair labeled as ⟨generation algorithm⟩-⟨number of nodes in each graph⟩. For BA graphs, we set the edge density to 4. For ER graphs, we set the edge density to 0.0016. For WS graphs, we set the rewiring probability to 0.2 and ring density to 4.

| Method | BA-10k | ER-10k | WS-10k |
|---|---|---|---|
| MCSP | 0.952 | 0.975 | 0.911 |
| MCSP+RL | 0.964 | 0.975 | 0.911 |
| GLSEARCH | **1.000** | **1.000** | **1.000** |
| BEST SOLUTION SIZE | 3172 | 2190 | 4839 |

## E.4  RESULTS ON GRAPH PAIRS WITH KNOWN MCS SIZE LOWER BOUND

To better understand the quality of subgraphs found by GLSEARCH, we construct new datasets with known lower bound MCS sizes. This is accomplished through generating 2 new graphs that share a common subgraph from the existing large real-world graph datasets. For each real-world graph, $\mathcal{G}_0$, we randomly extract 3 different subgraphs of the same size, $\mathcal{S}_0$, $\mathcal{S}_1$, and $\mathcal{S}_2$, by running breadth first search from 3 different starting nodes and extracting the explored induced subgraph. To construct a new graph pair, $(\mathcal{G}_1, \mathcal{G}_2)$, we form $\mathcal{G}_1$ by connecting $\mathcal{S}_0$ to $\mathcal{S}_1$ with 20 random edges and form $\mathcal{G}_2$ by connecting $\mathcal{S}_0$ to $\mathcal{S}_2$ with 20 random edges. Thus, connections between $\mathcal{S}_0$ nodes are the same in both $\mathcal{G}_1$ and $\mathcal{G}_2$, but connections between $\mathcal{G}_1 \setminus \mathcal{S}_0$ and $\mathcal{G}_2 \setminus \mathcal{S}_0$ are different. Notice, the lower bound of MCS size in these new datasets would be $|\mathcal{S}_0|$, and we name the new dataset by adding 'ss' to the parent dataset's name.

Table 10: Results on graph pairs with a common core subgraph (lower bound of MCS), with a fixed runtime of 10 minutes.

| Method | ROAD-ss | DBEN-ss | DBZH-ss | ENRO-ss | COPR-ss | HPPI-ss |
|---|---|---|---|---|---|---|
| | 444 | 778 | 762 | 1346 | 1406 | 860 |
| MCSP | 0.588 | 0.466 | 0.544 | 0.216 | 1.000 | 0.233 |
| MCSP+RL | 0.588 | 0.466 | 0.544 | 0.214 | 1.000 | 0.233 |
| GLSEARCH | **1.000** | **1.000** | **1.000** | **1.000** | **1.000** | **1.000** |
| BEST SOLUTION SIZE | 187 | 388 | 349 | 672 | 702 | 429 |
| CORE (LOWER BOUND) SIZE | 222 | 389 | 381 | 673 | 703 | 430 |

## F  EXTENSIONS OF GLSEARCH

GLSEARCH can be extended for a flurry of other MCS definitions, e.g. approximate MCS, MCS for weighted and directed graphs, etc. via a moderate amount of change to the search and learning components. In this section, we briefly outline what could be done for these tasks.

For approximate MCS detection, the bidomain constraint must be relaxed. One method of relaxing this constraint is to allow sets of nodes belonging to different but similar bidomains to match to each

other. For instance, nodes in $\mathcal{G}_1$ from the bidomain of bitstring "00110" could map with nodes in $\mathcal{G}_2$ from the bidomain of bitstring "00111", since they are only 1 hamming distance away. Such relaxations as this can be made stricter or looser based on the application. The difference would be the search framework, thus the learning part of GLSEARCH can largely stay the same.

Regarding MCS for graphs with non-negative edge weights, assuming our task is to maximize the sum of edge weights in the MCS, instead of defining $r_t = 1$, we can alter the reward function to be the difference of the sum of edge weights before and after selecting a node pair $r_t = \Sigma_{e \in S_t^{(u,v)}} w(e) -$

$\Sigma_{e \in S_t} w(e)$ where $S_t$ is the edges of currently selected subgraph, $S_t^{(u,v)}$ is the edges of the subgraph after adding node pair, $(u,v)$, and $w(\cdot)$ is a function that takes and edge and returns its weight. As the cumulative sum of rewards at step T is the sum of edge weights $\Sigma_{t \in [1,...,T]} r_t = \Sigma_{e \in S_T} w(e)$ and reinforcement learning aims to maximize the cumulative sum of rewards, we can adapt GLSearch to optimize for MCS problems with weighted edges.

Regarding MCS for directed graphs, the bidomain constraint may be altered such that every bit in the bidomain string representations now has 3 states: '0' for disconnected, '1' for connected by in-edge, and '2' for connected by out-edge. By considering the inward/outward direction of a bitstring, we can guarantee the isomorphism of directed graphs. In this case, the search framework would only differ in how bidomains are partitioned. The learning part of GLSearch would stay the same for this application.

More generally, we believe that there are many more extensions to GLSearch in addition to the ones listed, such as disconnected MCS, network alignment, or subgraph extraction. Further exploration of these are to be done as future efforts.

# G  RESULT VISUALIZATION

We plot the testing graph pairs and the results of MCSP and GLSEARCH in this Section. For all the figures except Figure 22, we use two colors for nodes, one for the selected subgraphs by the model, the other for the remaining subgraphs that cannot be further matched within the search budget. When plotting, we use larger circle size for nodes with larger degrees.

In general, GLSEARCH is a less interpretable but more powerful method compared to heuristic baselines. That said, GLSEARCH presents some insights that may be useful for producing new hand-crafted heuristics.

GLSEARCH identifies "smart" nodes which can lead to larger common subgraphs faster. For example, in the road networks (ROAD), as in Figure 10 and 11, our learned policy selects nodes with smaller degrees which allow for easier matching. The common subgraphs in road networks are most likely long chains, where nodes tend to have low degrees. In contrast MCSP always chooses high-degree nodes first leading to smaller extracted subgraphs.

GLSEARCH identifies "smart" *matching* of nodes which can lead to larger common subgraphs faster. For example, in the circuit graph (CIRC), we find 3 high-degree nodes that, when correctly matched, greatly reduces the matching difficulty of remaining nodes (see Figure 22 and 23). Upon further analysis, MCSP incorrectly matches the 3 high degree nodes (matching high degree node to low degree node). This happens when matching high-degree node correctly would break the isomorphism constraint (due to the current selected subgraph being incorrectly matched). GLSEARCH conscientiously adds node pairs so that it will always be able to match the 3 high degree nodes correctly.

We believe two aspects of GLSEARCH design lead to this phenomenon. First, GLSEARCH encodes neighborhood structures that are k-hop away. MCSP only looks at a single node and not its relationship with k-hop neighbors. Second, GLSEARCH considers scores on the node-node pair granularity, thus it will only match nodes with similar local neighborhoods. MCSP only considers scores on the node granularity, potentially matching 2 nodes with dissimilar neighborhoods together.

From these insights, one can potentially design a heuristic to first detect highly valuable nodes and guide a policy which prioritizes the matching of these critical nodes, or create better heuristics that consider not only uses the features of a single node but also the similarity between the 2 nodes being matched.

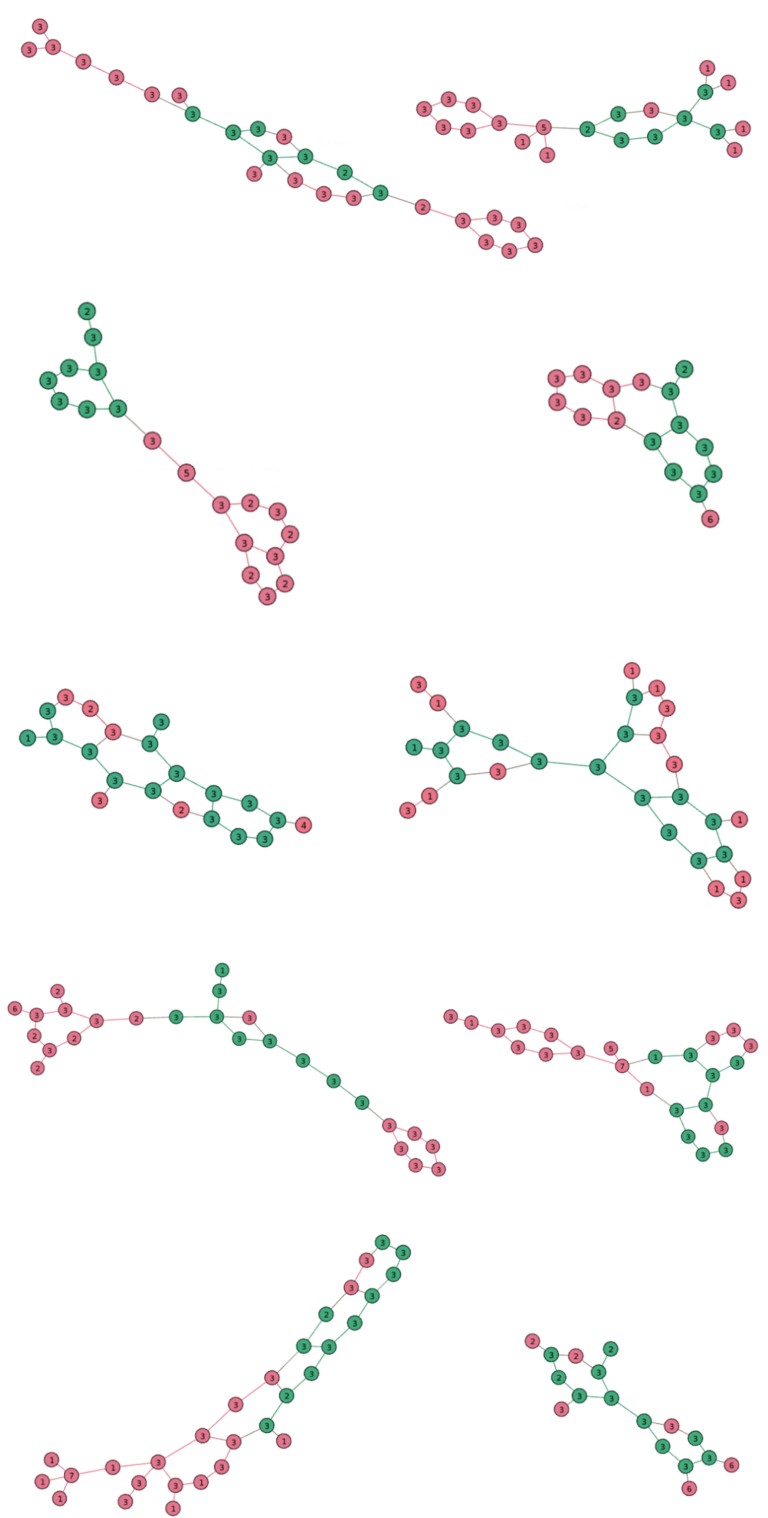

Figure 9: Visualization of 5 sampled graph pairs with the MCS results by GLSEARCH on NCI109. Each chemical compound node has its label indicated in the plot. Extracted subgraphs are highlighted in green.

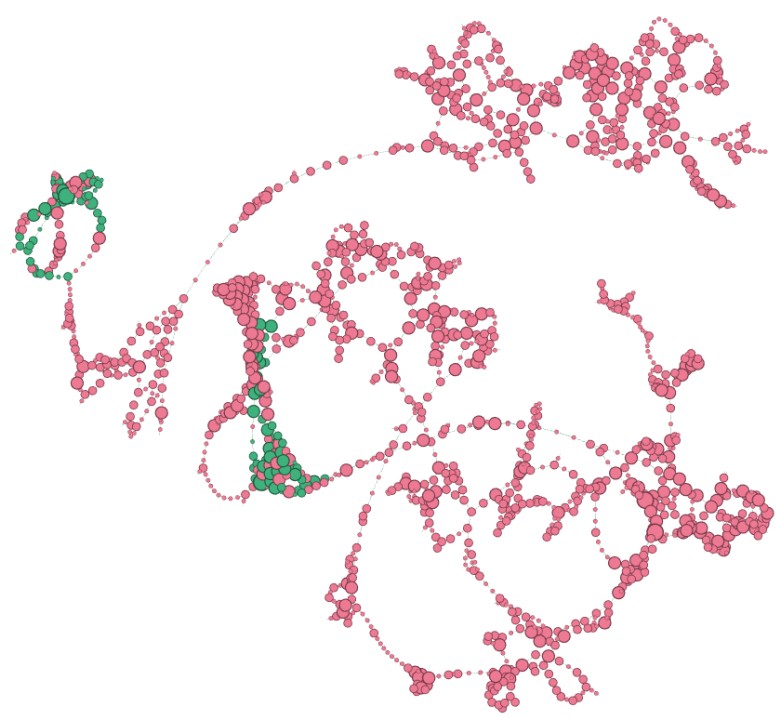

Figure 10: Visualization of MCSP result on ROAD. Extracted subgraphs are highlighted in green.

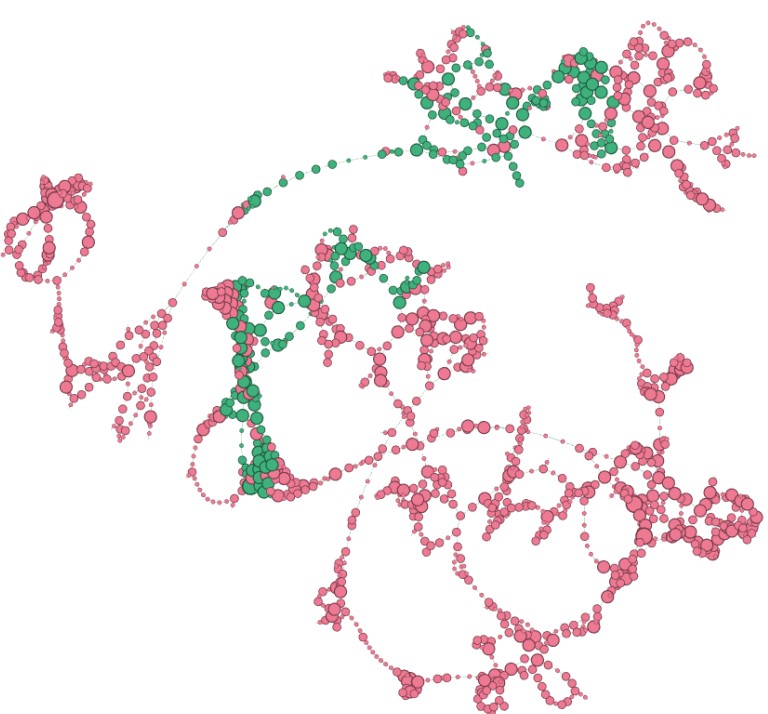

Figure 11: Visualization of GLSEARCH result on ROAD. Extracted subgraphs are highlighted in green.

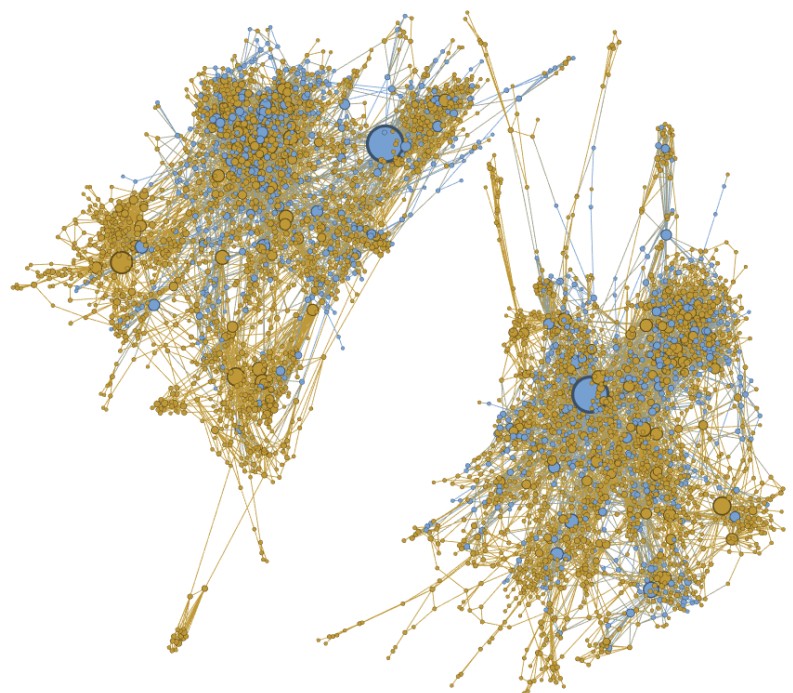

Figure 12: Visualization of MCSP result on DBEN. Extracted subgraphs are highlighted in blue.

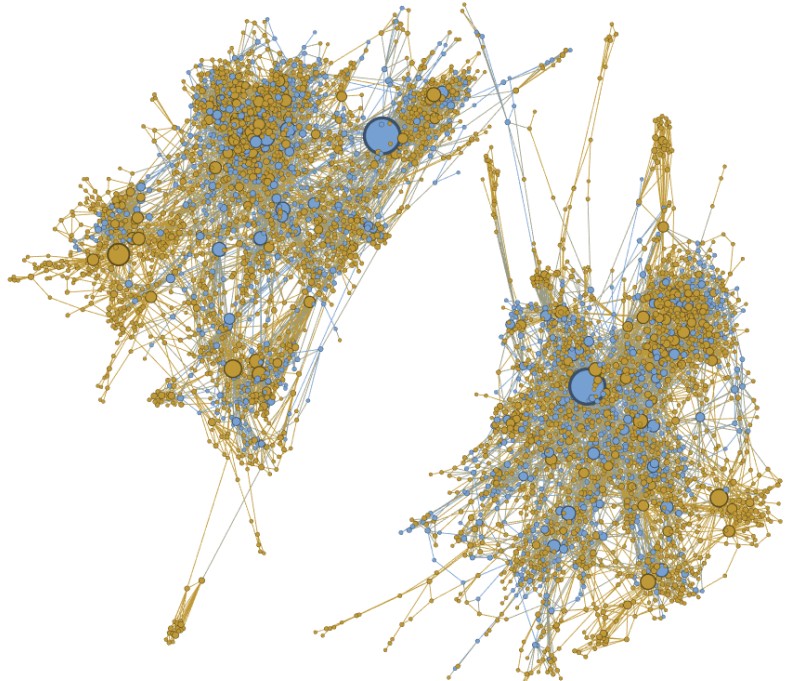

Figure 13: Visualization of GLSEARCH result on DBEN. Extracted subgraphs are highlighted in blue.

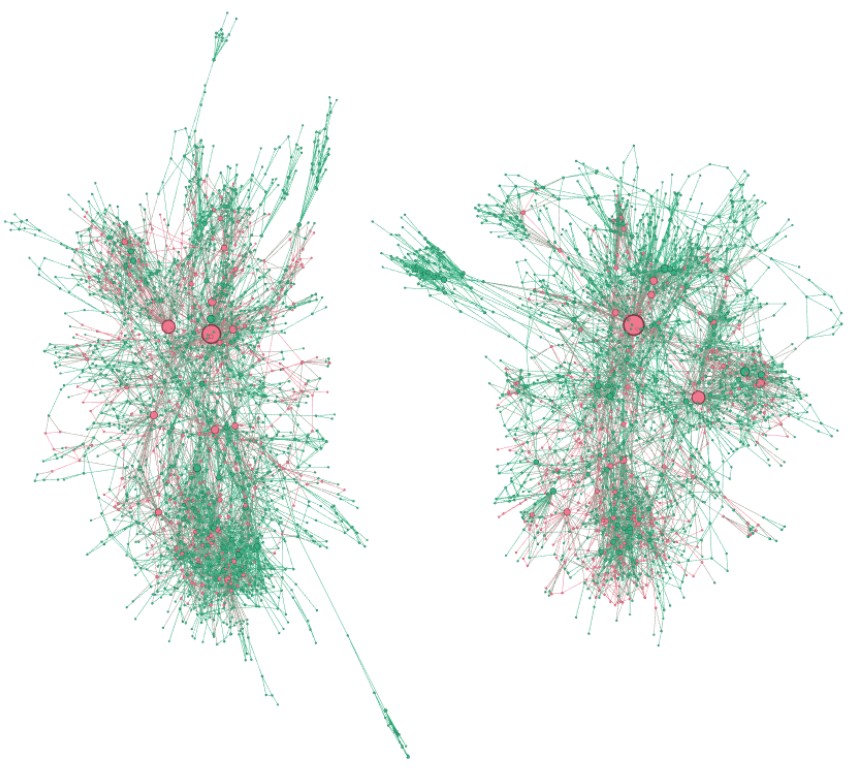

Figure 14: Visualization of MCSP result on DBZH. Extracted subgraphs are highlighted in pink.

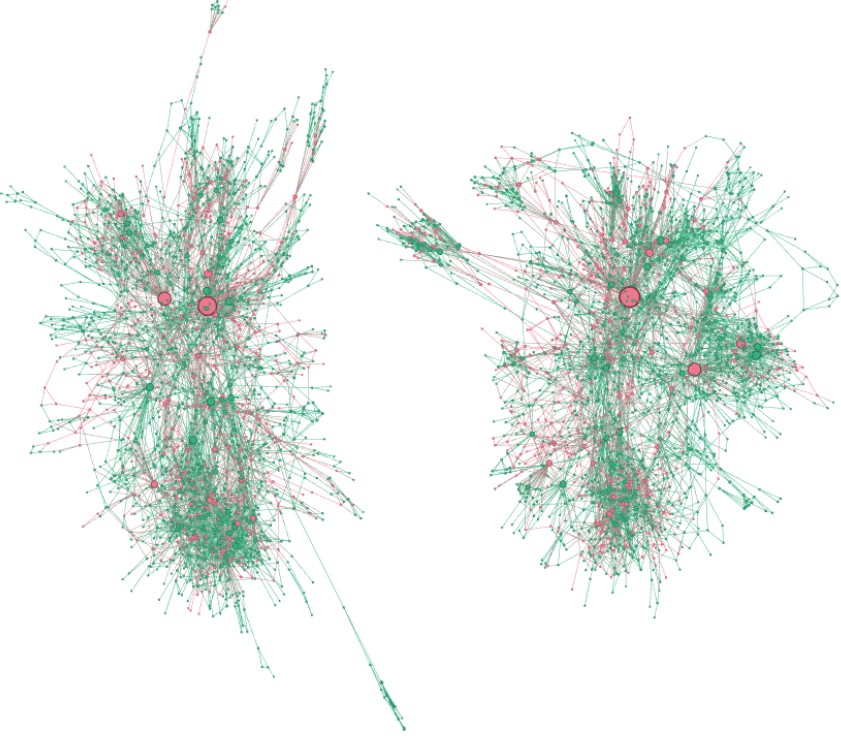

Figure 15: Visualization of GLSEARCH result on DBZH. Extracted subgraphs are highlighted in pink.

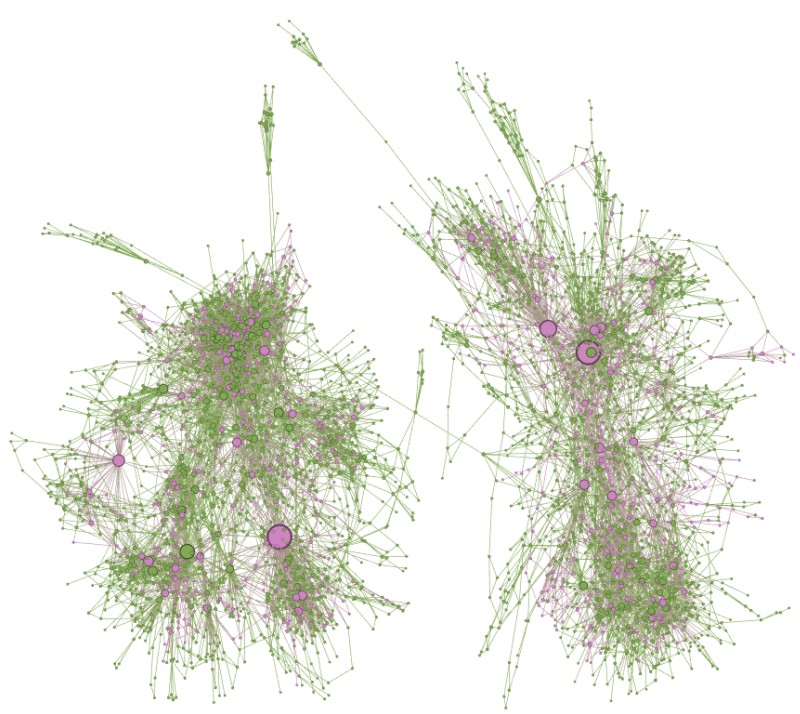

Figure 16: Visualization of MCSP result on DBPD. Extracted subgraphs are highlighted in purple.

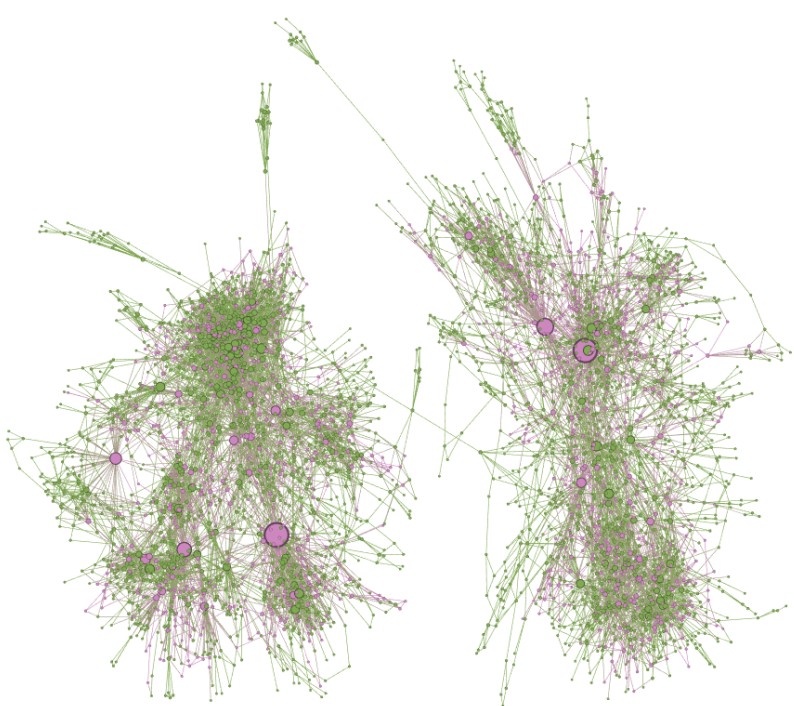

Figure 17: Visualization of GLSEARCH result on DBPD. Extracted subgraphs are highlighted in purple.

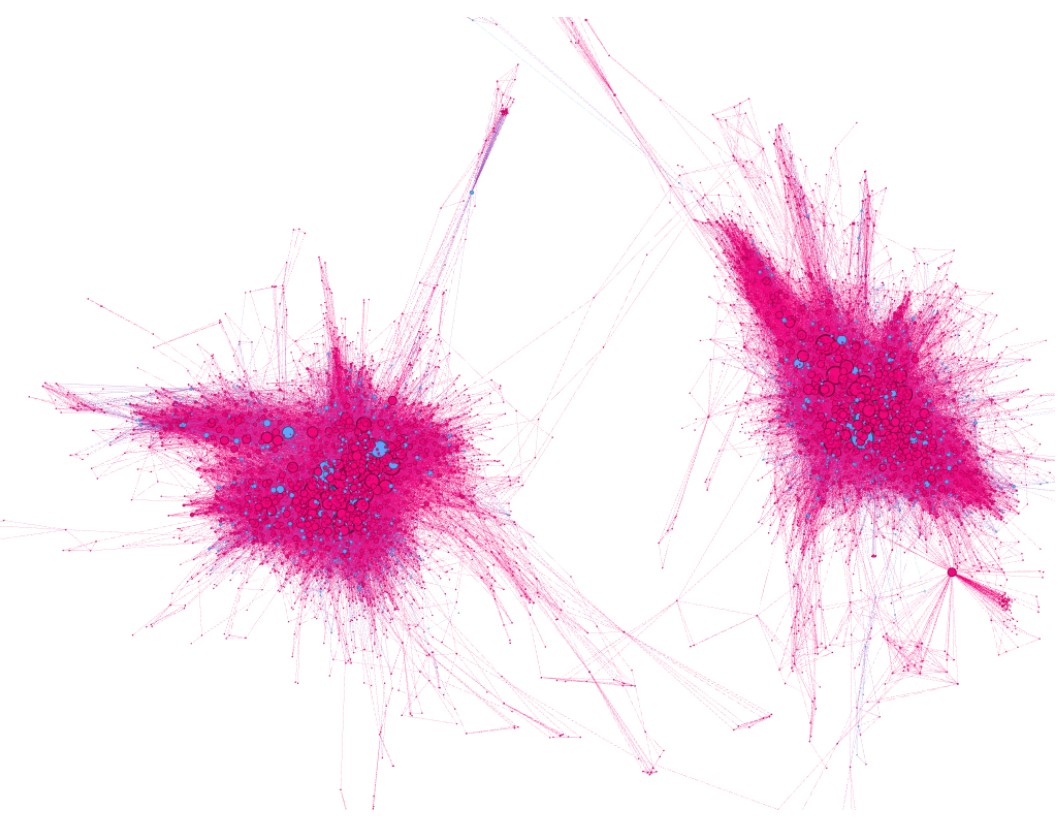

Figure 18: Visualization of MCSP result on ENRO. Extracted subgraphs are highlighted in blue.

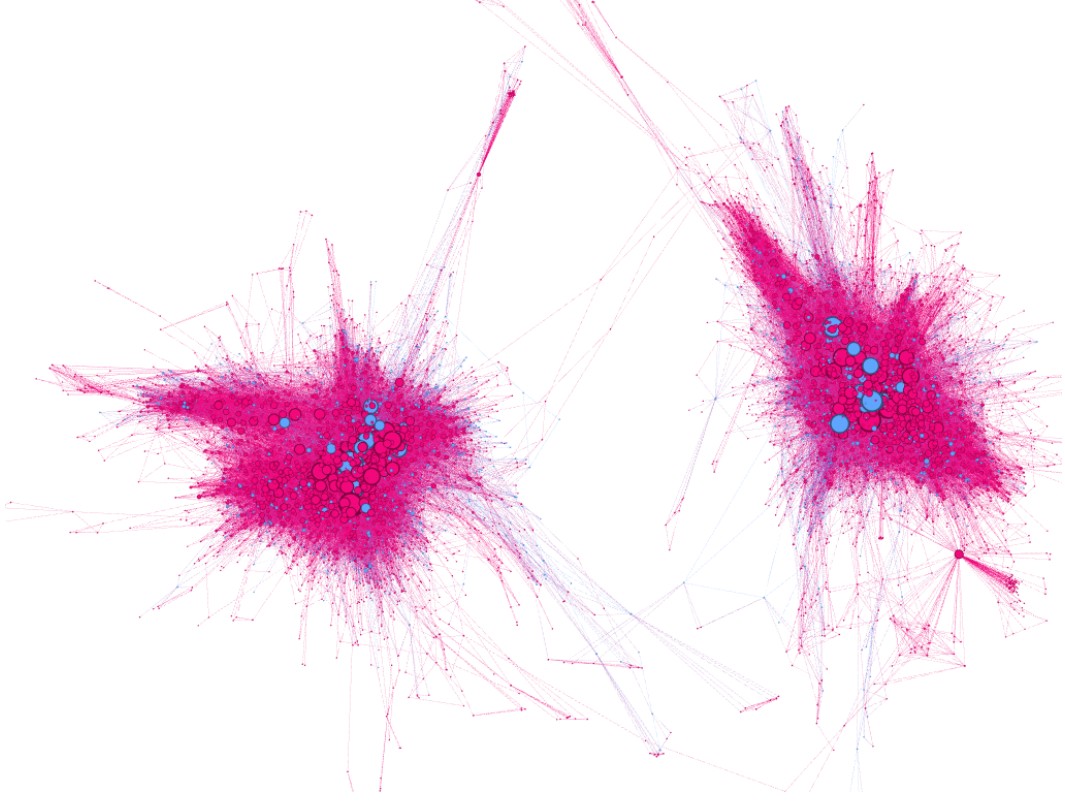

Figure 19: Visualization of GLSEARCH result on ENRO. Extracted subgraphs are highlighted in blue.

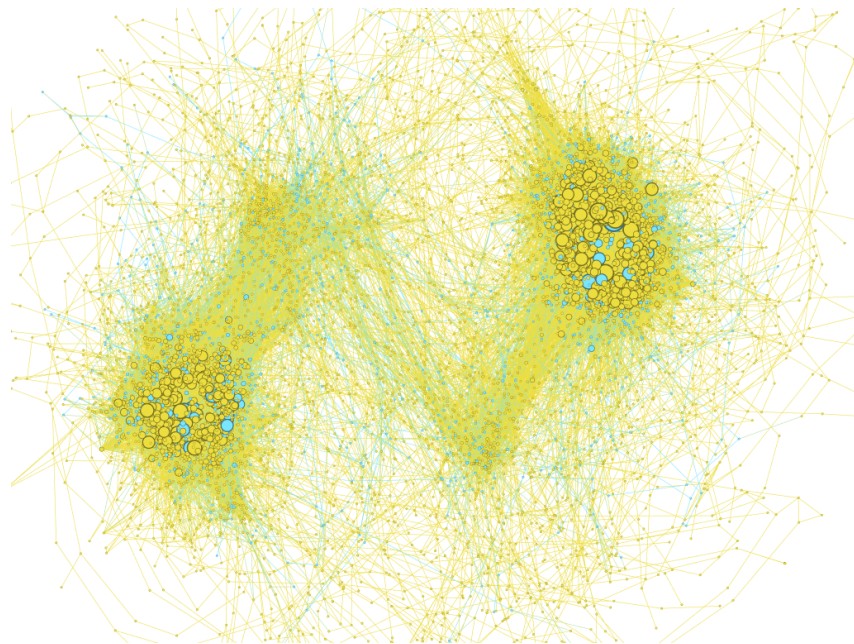

Figure 20: Visualization of MCSP result on COPR. Extracted subgraphs are highlighted in blue.

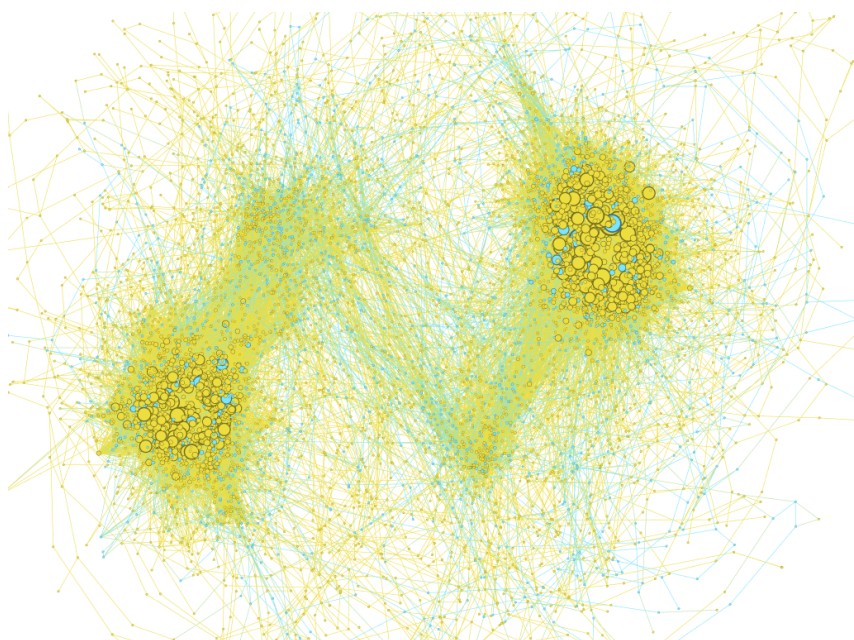

Figure 21: Visualization of GLSEARCH result on COPR. Extracted subgraphs are highlighted in blue.

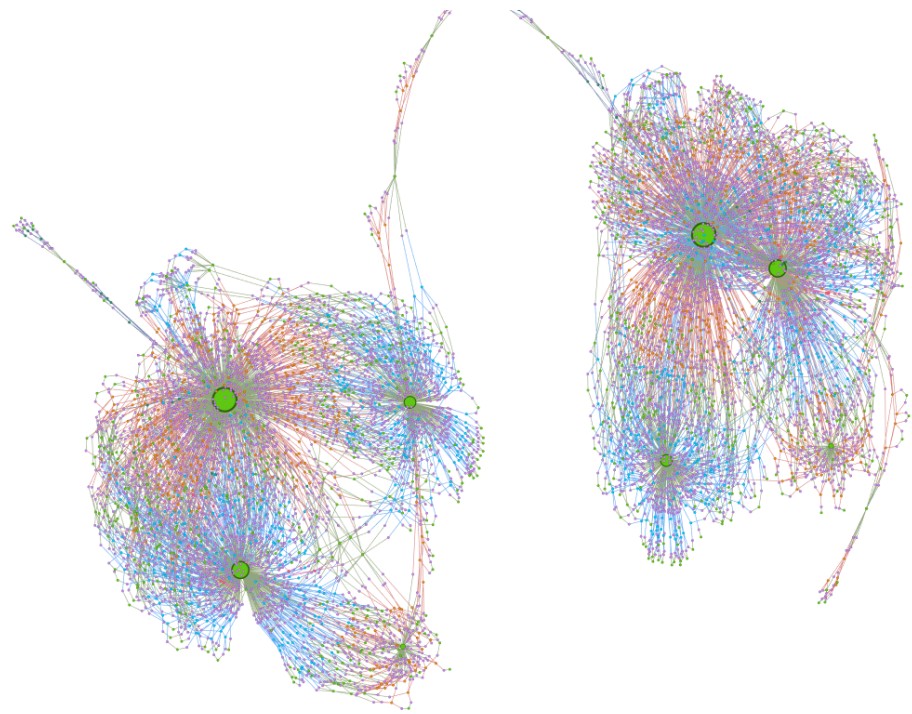

Figure 22: Visualization of the original graph pair of CIRC. Different colors denote different node labels. There are 6 node labels in total: M (71.67%), null (10.41%), PY (9.1%), NY (8.23%), N (0.37%), and P (0.21%).

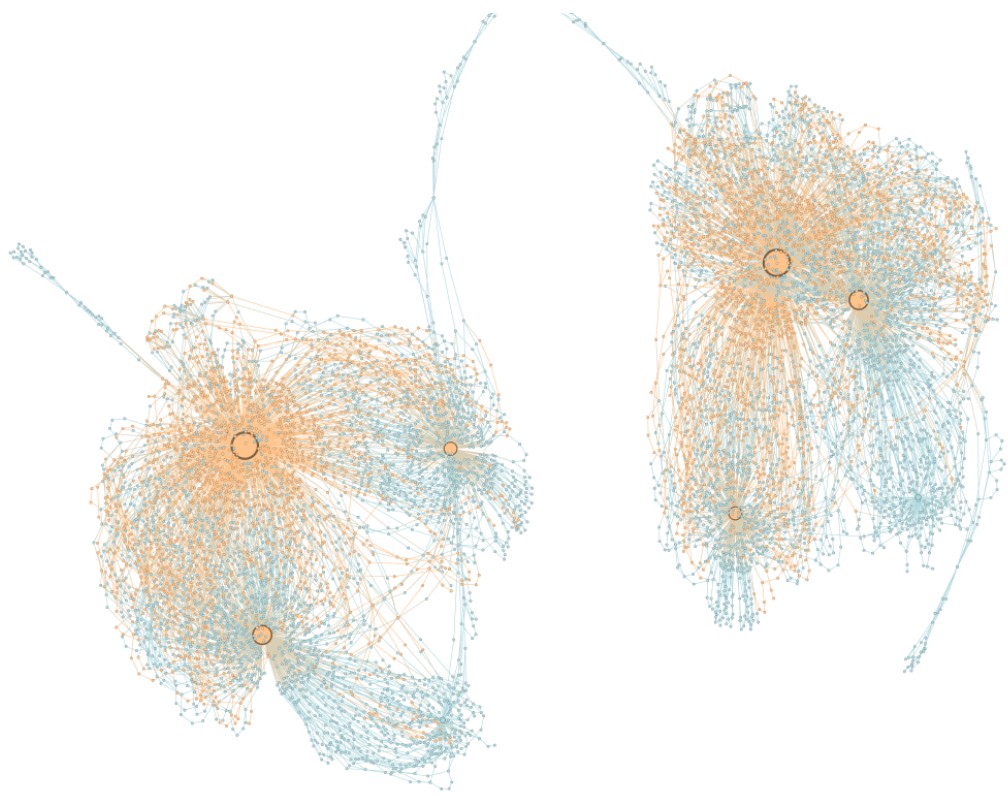

Figure 23: Visualization of MCSP result on CIRC. Extracted subgraphs are highlighted in yellow.

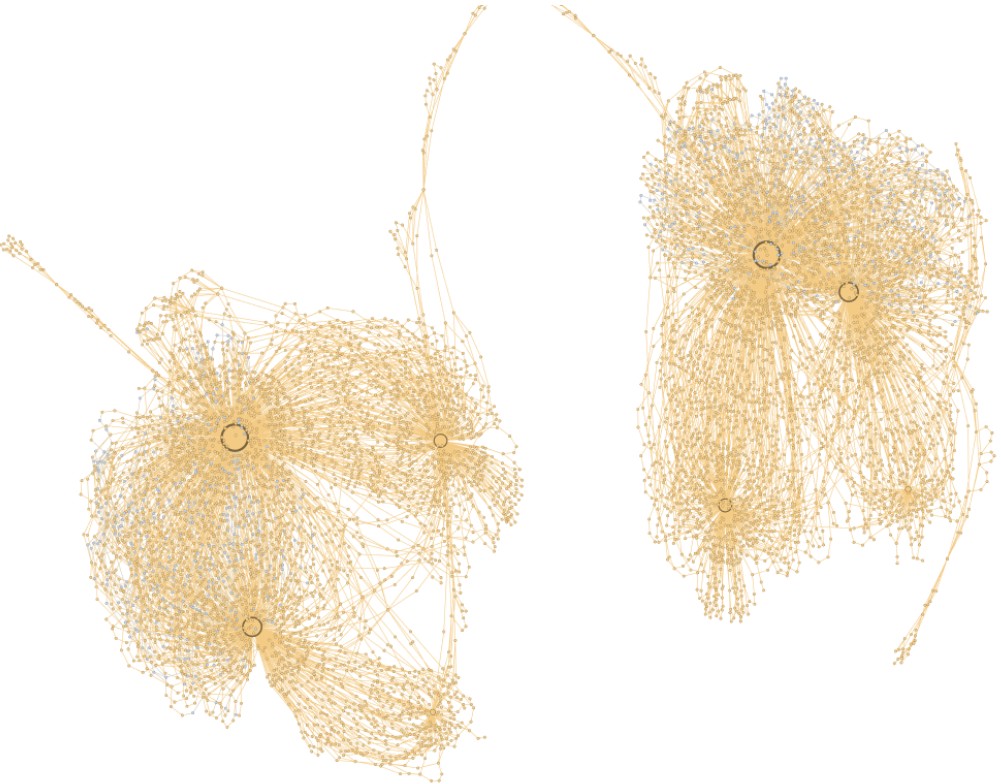

Figure 24: Visualization of GLSEARCH result on CIRC. Extracted subgraphs are highlighted in yellow.

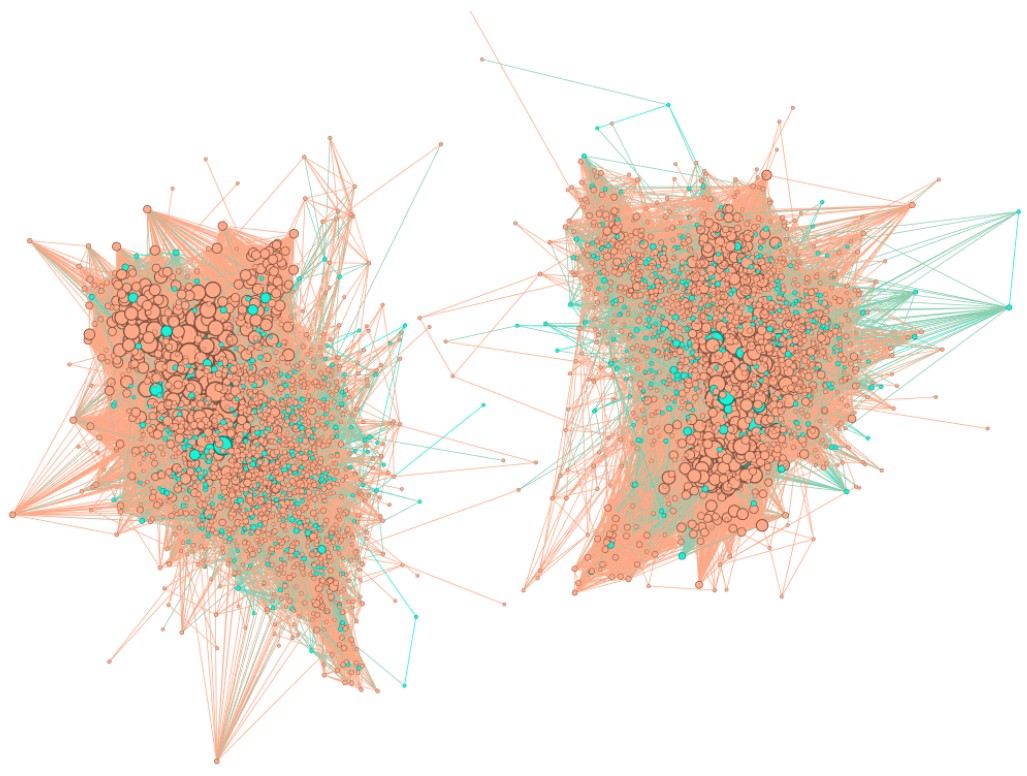

Figure 25: Visualization of MCSP result on HPPI. Extracted subgraphs are highlighted in cyan.

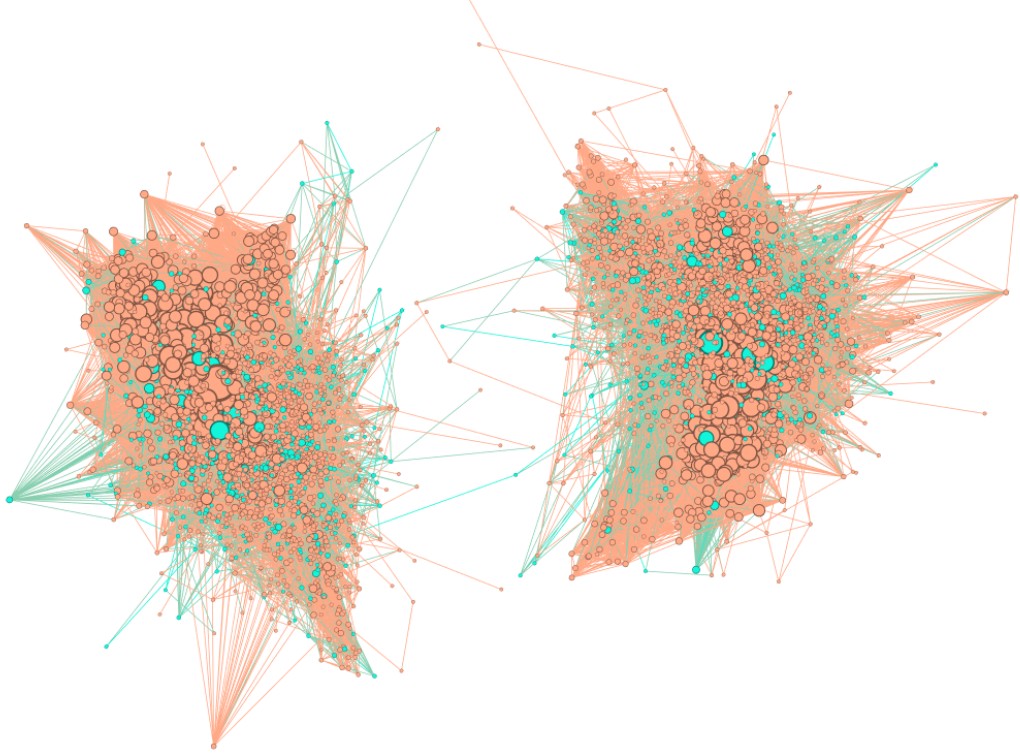

Figure 26: Visualization of GLSEARCH result on HPPI. Extracted subgraphs are highlighted in cyan.

