# OpenReview forum: "Learning to Search for Fast Maximum Common Subgraph Detection"
_ICLR.cc/2021/Conference — Reject_

### Official Review · AnonReviewer1 · 2020-10-29
**Bringing together RL and GNNs to guide search for MCS**

**Rating:** 5
**Confidence:** 3

**Review:**

Given two input graphs G1,G2 the maximum common subgraph detection problem asks to find an induced subgraph of both G1 and G2, with as many vertices as possible. In the recent years, there have been papers that introduce different heuristics for guiding the search of this subgraph within branch & bound algorithms. The main contribution of this paper is a combination of graph neural network embeddings and RL to guide the search more efficiently.  The function used to guide the deep Q-network is given in Equation (3). The paper performs a set of experiments on synthetic and real world pairs of graphs, where it is shown that it performs well in practice. The supplementary material provides more details on the experiments.

- While the exploration strategy is more sophisticated than previous works, it comes at a greater computational cost than prior work. The experimental section should explain the trade-offs. Some of the plots from the supplementary material should be included in the main text, but the overhead that one needs to pay due to running GNNs and RL should be clearly stated.
- In figure 2, shouldn't the output graphs, be induced, MCS subgraphs of the input graphs? While the figure serves the purpose of illustrating the exploration idea, it is a bit confusing. The caption and/or the text should better clarify the description of figure 2.
- Are there some interpretable heuristics that you can derive by studying the policy of your algorithm?
- It would be interesting to observe the effect of planted isomorphic subgraphs within larger graphs, with different connectivity patterns. For instance suppose we plant a large isomorphic subgraph on S={1..k} for convenience in two graphs G1,G2, but the connection between S, V/S is totally different between G1,G2. How would this affect GNN embedding for instance?
- Given that some times inputs are noisy, are you aware of works where the goal is to find approximate MCS subgraphs, i.e., a small number of edges differing between the two subgraphs? This would cause issues to the branch-and-bound policy, but nonetheless it is interesting to think whether your method can be adapted to this case within a different search framework? What about when graphs have (non-negative) weights on their edges, or they are directed? Have you performed experiments on the latter two settings?
- The authors motivate their problem with applications in drug discovery, chemoinformatics etc. Such an application is missing. Furthermore, for many real-world applications the input graphs have specific structure that enable the discovery of large common induced subgraphs even in polynomial time provably.  See for example "A polynomial-time maximum common subgraph algorithm for outerplanar graphs and its application to chemoinformatics" by Leander Schietgat, Jan Ramon & Maurice Bruynooghe. While this fact does not render the contributions of the author(s) useless, it does reduce their value in terms of application domains, that are not appropriately discussed in the paper.

Overall I found the paper to be interesting, but there are some non-trivial issues that need to be better addressed.

---

> ### Author Response · Authors · 2020-11-14
> **Response to 3**
>
> In general, GLSearch is a less interpretable but more powerful method compared to baselines. That said, we did find trends from GLSearch that may be useful to producing hand-crafted heuristics.
>
> GLSearch identifies “smart” nodes which can lead to larger common subgraphs faster. For example, in the road networks (“ROAD”), as in Figure 3 in the main text, our learned policy selects nodes with smaller degrees which allow for easier matching (The common subgraphs in road networks are most likely long chains, where nodes tend to have low degrees), while in contrast McSp always chooses high-degree nodes first leading to smaller extracted subgraphs.
>
> GLSearch identifies “smart” matching of nodes which can lead to larger common subgraphs faster. For example, in the circuit graph (“CIRC”), we find 3 high-degree nodes that, when correctly matched, greatly reduces the matching difficulty of remaining nodes (see Figure 23 and 24 in Supplementary Material). Upon further analysis, McSp incorrectly matches the 3 high degree nodes (matching high degree node to low degree node). This happens when matching high-degree node correctly would break the isomorphism constraint (due to the current selected subgraph being incorrectly matched). GLSearch conscientiously adds node pairs so that it will always be able to match the 3 high degree nodes correctly.
>
> We believe that 2 aspects of GLSearch design lead to this phenomenon. First, GLSearch encodes neighborhood structures that are k-hop away. McSp only looks at a single node and not its relationship with k-hop neighbors. Second, GLSearch considers scores on the node-node pair granularity, thus it will only match nodes with similar local neighborhoods. McSp only considers scores on the node granularity, potentially matching 2 dissimilar nodes together.
>
> From these insights, one can potentially design a heuristic to first detect highly valuable nodes and guide a policy which prioritizes the matching of these critical nodes, or create better heuristics that consider not only uses the features of a single node but also the similarity between the 2 nodes being matched.
>
> A common issue is these trends are somewhat hard to quantify universally. Learning based solutions often capture many such trends and automatically find the optimal node-pair selection policy, as is done by GLSearch.
>
> This is now discussed in Section G of our latest submission.

---

> ### Author Response · Authors · 2020-11-14
> **Response to 2**
>
> We apologize for the confusion caused by the description of Figure 2. In Figure 2, the extracted subgraphs are drawn for states 4, 5, 6, 7, 11, 12, 13, and 14. Regarding the output graphs: We maintain the best solution found so far during search, so at the end of search, the output graphs would correspond to state 13 or 14 (since the subgraph sizes are the same and the largest, 5). According to the definition of induced subgraphs (Section 2.1), the subgraphs corresponding to state 13 (or 14) satisfy the induced constraint, i.e. all edges between selected subgraph nodes in the whole graph are also included in the subgraph. Please let us know any further concerns.

---

> ### Author Response · Authors · 2020-11-15
> **Response to 1: Trade-off between overhead and performance: Our model is faster AND more accurate**
>
> In order to see the effect of the overhead incurred by the neural network operations, instead of giving a fixed search iteration budget like in Table 2 of the main text, we set a fixed running time budget of 600 seconds (in wall time), and compare GLSEARCH against baseline methods:
>
> ````````````~~~
>               | Road  | DbEn  | DbZh  | Dbpd  | Enro  | CoPr  | Circ  | HPpi
> Num Nodes     | 652   | 1945  | 1907  | 1907  | 3369  | 3518  | 4275  | 2152
> --------------+-------+-------+-------+-------+-------+-------+-------+------
> McSp          | 0.357 | 0.819 | 0.800 | 0.725 | 0.696 | 0.684 | 0.432 | 0.866
> McSp+RL       | 0.921 | 0.819 | 0.800 | 0.725 | 0.696 | 0.684 | 0.431 | 0.866
> GW-QAP        | 0.486 | 0.949 | 0.879 | 0.815 | 0.710 | 0.868 | 0.308 | 0.836
> I-PCA         | 0.257 | 0.554 | 0.555 | 0.612 | 0.655 | 0.723 | 0.145 | 0.762
> NMcs          | 0.993 | 0.787 | 0.615 | 0.619 | 0.736 | 0.738 | 0.095 | 0.784
> GLSearch-Rand | 0.679 | 0.744 | 0.703 | 0.675 | 0.686 | 0.702 | 0.225 | 0.836
> GLSearch      | 1.000 | 1.000 | 1.000 | 1.000 | 1.000 | 1.000 | 1.000 | 1.000
> --------------+-------+-------+-------+-------+-------+-------+-------+--------
> Best Sol Size | 140   | 507   | 481   | 520   | 542   | 794   | 4060  | 403
> ````````````~`~~
>
> Average Running Time Per Iteration [ms]
> ````````````~~~
>               | Road   | DbEn   | DbZh   | Dbpd   | Enro    | CoPr   | Circ    | HPpi
> --------------+--------+--------+--------+--------+---------+--------+---------+-------
> McSp          | 2.040  | 10.724 | 1.415  | 0.974  | 1.722   | 2.891  | 1.776   | 0.498
> McSp+RL       | 0.894  | 6.834  | 2.103  | 1.247  | 2.166   | 3.107  | 2.080   | 0.559
> GW-QAP        | 0.548  | 0.834  | 0.546  | 4.692  | 1.041   | 3.419  | 1.550   | 0.546
> I-PCA         | 1.152  | 1.797  | 0.967  | 0.897  | 1.739   | 2.725  | 3.792   | 0.636
> NMcs          | 2.394  | 4.172  | 4.648  | 5.667  | 9.610   | 9.788  | 7.471   | 15.742
> GLSearch-Rand | 17.392 | 66.418 | 67.342 | 67.946 | 163.005 | 71.972 | 655.447 | 83.488
> GLSearch      | 8.132  | 66.552 | 71.409 | 96.262 | 135.087 | 51.181 | 37.377  | 60.509
> ````````````~`~~
>
> Number of Iterations:
> ````````````~~~
>               | Road    | DbEn   | DbZh    | Dbpd   | Enro   | CoPr   | Circ   | HPpi
> --------------+---------+--------+---------+--------+--------+--------+--------+--------
> McSp          | 294090  | 55951  | 424114  | 615973 | 348367 | 207564 | 337836 | 1204233
> McSp+RL       | 671221  | 87794  | 285358  | 481087 | 276975 | 193119 | 288498 | 1080304
> GW-QAP        | 1094383 | 727762 | 1098605 | 127870 | 576256 | 175506 | 387128 | 1098047
> I-PCA         | 520749  | 333797 | 620790  | 668599 | 345073 | 220168 | 158245 | 943475
> NMcs          | 250639  | 143833 | 129082  | 105886 | 62433  | 61301  | 80311  | 38116
> GLSearch-Rand | 34508   | 9041   | 8910    | 8833   | 3683   | 8340   | 917    | 7187
> GLSearch      | 73784   | 9016   | 8469    | 6242   | 4442   | 11723  | 16090  | 9949
> ````````````~`~~
>
> Combined with the average running time per iteration, it is clear that although GLSEARCH spends more time per iteration due to the DQN computation, for the entire search the additional running time is well worth because after the same amount of time GLSEARCH finds larger solutions due to “smarter” decisions made during the search. The trade-off between computational overhead and performance in our case is not an issue -- overall GLSEARCH finds better MCS solutions in fewer iterations AND less running time.
>
> Please refer to Section 4.5 and Section E.1 of our latest submission for detailed plots.

---

> ### Author Response · Authors · 2020-11-17
> **Response to 4**
>
> To investigate the effect of planting isomorphic subgraphs within larger graphs, we construct new datasets from our existing large real-world graphs. For each real-world graph, G0, we randomly extract 3 different subgraphs of the same size, S0 (core), S1, and S2, through running BFS at 3 different starting nodes then extracting the explored nodes with induced edges. From these 3 subgraphs, we form G1 by connecting S0 to S1 with 20 random edges and form G2 by connecting S0 to S2 with 20 random edges. In other words, connections between S0 nodes are the same in both G1 and G2, but connections between V1/S0 and V2/S0 are different, where V1 and V2 are G1’s and G2’s nodes respectively. G1 and G2 form the pair for the new dataset. We denote this new dataset by adding ‘-ss’ to the parent dataset’s name.
>
> ````````````~~~
>                 | Road-ss | DbEn-ss | DbZh-ss | Enro-ss | CoPr-ss | HPpi-ss
> ----------------+---------+---------+---------+---------+---------+--------
> McSp            | 0.588   | 0.466   | 0.544   | 0.216   | 1.000   | 0.233
> McSp+RL         | 0.588   | 0.466   | 0.544   | 0.214   | 1.000   | 0.233
> GLSearch        | 1.000   | 1.000   | 1.000   | 1.000   | 1.000   | 1.000
> ----------------+---------+---------+---------+---------+---------+--------
> Best Sol Size   | 187     | 388     | 349     | 672     | 702     | 429
> S0 (Core) Size  | 222     | 389     | 381     | 673     | 703     | 430
> ````````````~`~~
>
> (Note: Dbpd consists of DbEn and DbZh graphs, thus constructing Dbpd-ss would result in either the DbEn-ss or DbZh-ss dataset, depending on which G0 we choose from Dbpd.)
>
> As you can see from the last two rows of the table, under all the settings, GLSearch can extract isomorphic common subgraphs that are close to the core subgraphs, indicating the superb ability of GLSearch to extract MCS. In contrast, McSp and McSp+RL usually can only find a small fraction of the core subgraphs.
>
> Section E.4 of the latest submission also shows the above results.

---

> ### Author Response · Authors · 2020-11-17
> **Response to 5**
>
> We have explored various other works to adapt our model for other MCS related tasks. GLSearch can be extended for approximate MCS, MCS for weighted and directed graphs, etc. via a moderate amount of change to the search and learning components.
>
> For approximate MCS, the bidomain constraint must be relaxed. One method of relaxing this constraint is to allow sets of nodes belonging to different but similar bidomains to match to each other. For instance, nodes in G1 from the bidomain of bitstring ‘00110’ could map with nodes in G2 from the bidomain of bitstring ‘00111’, since they are only 1 hamming distance away. Such relaxations as this can be made stricter or looser based on the application. As you pointed out, the difference would be the search framework, thus the learning part of GLSearch can largely stay the same.
>
> Regarding MCS for graphs with non-negative edge weights, assuming our task is to maximize the sum of edge weights in the MCS, instead of defining $r_t=1$, we can alter the reward function to be the difference of the sum of edge weights before and after selecting a node pair $r_t = \Sigma_{e \in S_t^{(u,v)}} w(e) - \Sigma_{e \in S_t} w(e)$ where $S_t$ is the edges of currently selected subgraph, $S_t^{(u,v)}$ is the edges of the subgraph after adding node pair, $(u,v)$, and $w(\cdot)$ is a function that takes and edge and returns its weight. As the cumulative sum of rewards at step T is the sum of edge weights $\Sigma_{t \in [1,...,T]} r_t = \Sigma_{e \in S_T} w(e)$ and reinforcement learning aims to maximize the cumulative sum of rewards, we can adapt GLSearch to optimize for MCS problems with weighted edges.
>
> Regarding MCS for directed graphs, the bidomain constraint may be altered such that every bit in the bidomain string representations now has 3 states: ‘0’ for disconnected, ‘1’ for connected by in-edge, and ‘2’ for connected by out-edge. By considering the inward/outward direction of a bitstring, we can guarantee the isomorphism of directed graphs. In this case, the search framework would only differ in how bidomains are partitioned. The learning part of GLSearch would stay the same for this application.
>
> More generally, we believe that there are many more extensions to GLSearch in addition to the ones listed, such as disconnected MCS, network alignment, or subgraph extraction. We plan to explore these in the future, and thank you very much for the interesting extensions you described!
>
> Section F of the latest submission talks about this topic.

---

> ### Author Response · Authors · 2020-11-17
> **Response to 6**
>
> We have added a new dataset with 100 graph pairs from the NCI109 dataset, a dataset of labeled chemical compounds, and ran all baselines for 500 search iterations, which is consistent with other datasets in Table 1 of the main text.
>
> ````````````~~~
> Method    	| Avg Subgraph Size
> --------------+--------------------
> McSp      	| 0.948
> McSp+RL   	| 0.948
> GW-QAP    	| 0.966
> I-PCA     	| 0.951
> NMcs      	| 0.954
> GLSearch-RAND | 0.989
> GLSearch  	| 1.000
> --------------+--------------------
> Best Sol Size | 10.48
>  ````````````~`~~
>
> Since the chemical compounds are relatively small compared to the other datasets, all the methods perform similarly, yet our method still achieves the best results.
>
> We agree that incorporating domain knowledge about chemicals may further improve performance, and would leave incorporating human knowledge on specific application areas to future work.
>
> That being said, our technique also carries its merit. It is a very general framework, which does not rely on human heuristics and can automatically learn a good search strategy. This becomes very flexible when coming to a new domain or a new MCS definition where human knowledge is scarce. In particular, whenever heuristic driven search strategies are employed, there is potential to use GLSearch to further boost performance.
>
> We have updated our PDF submission with the new results in Table 1 with visualizations in Figure 8 in the supplementary material.

---

### Official Review · AnonReviewer2 · 2020-10-29
**This paper proposes GLSEARCH, a Graph Neural Network based model for MCS detection, which aims to search for the maximum common subgraphs between two input graphs.**

**Rating:** 5
**Confidence:** 4

**Review:**

The motivation of this paper is clear and interesting, as it’s important to explore the maximum common subgraph in biochemical domain. In this paper, the authors conduct a lot of experiments to demonstrate the effectiveness of the proposed method. Despite of this, the presentation of this paper requires improvement because many important details are missing, which makes it hard to follow. The time-complexity analysis might also be crucial to demonstrate the superiority of the proposed method over other baselines in terms of searching time.

Strengths:
1.	The motivation of this paper is clear and interesting.
2.	The authors conduct many experiments to demonstrate the effectiveness of the proposed method.
Weakness:
1.	The last paragraph in Section 2.2 (the notion of bidomain) is hard to follow. It’s not clear what is k, and how bidomain partitions the nodes to get V’_{k,1} and V’_{k,2}, which from two different graphs G_1 and G_2. Many details regarding the notations are missing when a new equation is introduced, e.g. r_t in Factoring Out Action subsection. These missing details make it hard to follow.
2.	In Figure 1, what’s the difference between 01 and 10?
3.	In equation 2, what operation does INTERACT stand for?
4.	The authors mention the maximum common subgraph detection problem is NP-hard, so it’s important to provide the time complexity of the proposed algorithms. However, in this paper, the authors do not provide any time complexity analysis or report the running time of the proposed method. In addition, in this paper, the authors mention that MCSP and MCSP+RL adopt the heuristics node pair selection policy but the proposed method is “learn-to-search” algorithm. It might be interesting to see whether the proposed method greatly reduces the search time compared with state-of-the-art algorithms as the number of nodes increases.

---

> ### Author Response · Authors · 2020-11-13
> **Response to 3 and 4**
>
> Thank you for your comments. Regarding the notion of bidomain: We would like to clarify that bidomains are labels associated with each node denoting its connectivity to the existing subgraph. By only matching nodes with the same connectivity pattern, we guarantee isomorphism. k is the index of bidomain at a particular state, so $D_k$ is the k-th bidomain. For example, in Figure 1, there are 3 bidomains, which we denote as $D_0$ (yellow), $D_1$ (green), and $D_2$ (purple).
>
> Each bidomain can be also represented as a bit string where bit is 1 if every node in this bidomain is connected to an already selected node and 0 if not connected to that selected node. For example, in Fig. 1, each node in the “10” bidomain is connected to the top “C” node in the subgraph and disconnected to the bottom “C” node in the subgraph; and each node in the “01” biodomain is connected to the bottom “C” and disconnected to the top “C”.
>
> Regarding $r_t$ in the Factoring Our Action subsection: In the formulation, $r_t$ is the immediate reward at timestep t, which is +1 since for each transition, one more node is selected in each of the two input graphs. This is mentioned at the end of the first paragraph of Section 3.1, and we apologize for missing the definition $r_t = 1$.

---

> ### Author Response · Authors · 2020-11-13
> **Response to 5**
>
> Regarding the INTERACT notation in Equation 2: INTERACT is a function that combines 2 embeddings in a commutative way (INTRERACT(a,b) = INTERACT(b,a)). This property is necessary to make DQN invariant from swapping the order of the 2 graphs. Suitable candidates for INTERACT include “+” or “max”. In our method, we use a 1D CNN followed by MLP whose details can be found in Section B.3 of Supplementary Material. Section 3.2 of the latest submission has been updated to clarify this.

---

> ### Author Response · Authors · 2020-11-13
> **Response to 6**
>
> Regarding time complexity: Overall the branch-and-bound search has exponential worst-case time complexity due to the NP-hard nature of exact MCS detection, and our goal is to use additional overhead per search iteration to make “smarter” decision each iteration so that we can find a larger common subgraph faster (in less iterations AND real running time). Per iteration, our model admittedly requires the neural network operations to compute a Q score instead of simply using a degree heuristic which is $O(1)$. Here we analyze the time complexity of these neural operations:
>
> 1 To compute the node embeddings, the complexity is the same as the GNN model, which in our case is $O(|V|+|E|)$ for GAT (since nodes must aggregate embeddings from neighbors and attention scores must be computed for each edge). Notice the node embeddings are computed by local neighborhood aggregation, and will not be updated in search, and therefore we compute the node embeddings only once at the beginning of search, and can be cached for efficiency.
>
> 2 At each iteration, to compute a Q score for a state-action pair, we run Equation 3 which requires computing the whole-graph, subgraph, and bidomain embeddings. Overall the time complexity is $O(|V|-|V_s|)$ where $|V_s|$ is the number of nodes in the currently matched subgraph. Please see the detailed analysis below:
>
> 2.1. The whole-graph embeddings do not change across search, so they only need to be computed once at the beginning.
>
> 2.2. The subgraph embeddings can be maintained incrementally, i.e. adding new node embeddings as search grows the subgraph.
>
> 2.3. The bidomain embeddings are computed via a series of READOUT and INTERACT operations (Equation 2).
>
> 2.3.1. For READOUT: We use summation followed by MLP so the runtime is $O(|V|-|V_s|)$.
>
> 2.3.2. For INTERACT: We use a 1D CNN followed by MLP which depends on the embedding dimension set to a constant, and does not depend on the number of nodes in the input graphs.
>
> Section C.7 in the latest submission discusses this aspect.
>
> To see how GLSearch reduces search time compared with state-of-the-art algorithms as the number of nodes increases, we construct 6 new datasets from the circuit dataset (CIRC) by sampling isomorphic subgraphs with differing numbers of nodes. In this case, we know the ground truth MCS, which is the whole graph. We set a fixed time budget of 10 minutes to find that our method outperforms heuristic baselines in all experiments, significant performance advantage (~2x performance) begins once graphs are larger than 256 nodes, and our model consistently matches almost the entire graph (Table following the same format as Figure 2 of main text):
>
> ````````````~~~
>      	     | Circ128 | Circ256 | Circ512 | Circ1024 | Circ2048 | Circ4096
> --------------+---------+---------+---------+----------+----------+---------
> McSp 	     | 0.812   | 0.559   | 0.279   | 0.282	| 0.480	| 0.638
> McSp+RL       | 0.812   | 0.559   | 0.268   | 0.432	| 0.476	| 0.625
> GLSearch      | 1.000   | 1.000   | 1.000   | 1.000	| 1.000	| 1.000
> --------------+---------+---------+---------+----------+----------+---------
> Best Sol Size | 128 	| 256 	| 512 	| 999  	| 2040 	| 4082
> ````````````~`~~

---

### Official Review · AnonReviewer5 · 2020-11-06
**Learning-based formulation of the MCS problem. Interesting approach based on GNNs and RL. Experiments on larger graphs would be interesting to consider.**

**Rating:** 7
**Confidence:** 4

**Review:**

The paper deals with the problem of Maximum Common Subgraph (MCS) detection, following a learning-based approach. In particular, it introduces GLSEARCH, a model that leverages representations learned by GNNs in a reinforcement learning framework to allow for efficient search. The proposed model has been experimentally evaluated on both artificial and real-world graphs, and its performance has been compared against traditional and learning-based baselines.

Strong points:

--- The paper deals with an important problem, and the overall learning-based formulation and solution look very interesting.

--- The paper is well-written and most concepts, especially the proposed approach, have been clearly presented. Besides, the supplementary material describes in detail most of the aspects of the paper.

--- The ablation study is interesting and demonstrates that the chosen architecture of GLSEARCH has consistent behavior.


Weak points:

--- My main concern is related to various aspects of the experimental evaluation of the proposed model. First, most of the datasets used in the evaluation seem to be unlabeled. In the basic formulation of the model though, the input graphs are allowed to be labeled. To my view, this makes the overall task more challenging. How consistent are the results in the case of labeled graphs?

--- Second, the size of the input graphs is also an important parameter. Definitely, most heuristic baselines might not be able to scale to graphs with more than a few thousands of nodes, but I would be expecting to consider some large-scale network containing a few tens of thousands of nodes for the evaluation of GLSEARCH.


Typos:

--- Page 4, first paragraph: *F*or MCS, …
--- Caption of Table 3: *Ablation*

---

> ### Author Response · Authors · 2020-11-18
> **Response to 1 about labeled graphs**
>
> Thank you for your feedback. We would like to point out that we have added a new labeled dataset NCI109 consisting of 100 graph pairs where each graph is a chemical compound whose nodes are labeled. The results are shown in Table 1 of the new uploaded PDF. In addition to the circuit dataset in Table 2 (CIRC), we have now 2 labeled datasets.
>
> It turns out that adding labels simplifies the task, because labels significantly reduce the amount of candidate node-node mappings at each state in the search. Concretely, we apply the label constraint as follows: at the start of search, add another bit (since node labels are discrete) to the bidomain bitstring representation indicating the label of the node. Because we can only match nodes within the same bidomain, we see the label constraint essentially creates more fine-grained bidomains, reducing the amount of candidate node-node pairs and thus decreasing difficulty of the task.

---

> ### Author Response · Authors · 2020-11-18
> **Response to 2 on scalability**
>
> Thank you for the encouraging comments. We still outperform baselines on large-scale networks containing tens of thousands of nodes. To verify this claim, we generated 6 large synthetic graphs using the Barabasi-Albert, Erdos-Renyi, and Watts-Strogatz algorithms, each with 10000 nodes, and ran GLSearch, McSp, and McSp+RL for 7500 iterations, and find our method still outperforms baselines (same table format as Figure 2 of main text):
>
> ````````````~~~
>           	| BA10000 | ER10000 | WS10000
> --------------+---------+---------+--------
> McSp 	     | 0.952   | 0.975   | 0.911
> McSp+RL       | 0.964   | 0.975   | 0.911
> GLSearch      | 1.000   | 1.000   | 1.000
> --------------+---------+---------+--------
> Best Sol Size | 3172	| 2190	| 4839
> ````````````~`~~
>
>
> We notice that GLSearch will retain consistent performance as graph size increases, but heuristic baselines will start to fail. This is validated by constructing 6 new datasets from the circuit dataset in Table 2 (CIRC) by sampling isomorphic subgraphs with differing number of nodes. In this case, we know the ground truth MCS, which is the whole graph. We run GLSearch, McSp, and McSp+RL with 7500 iterations, and show the results below (same table format as Figure 2 of main text):
>
> ````````````~~~
>           	| Circ128 | Circ256 | Circ512 | Circ1024 | Circ2048 | Circ4096
> --------------+---------+---------+---------+----------+----------+---------
> McSp      	| 0.812   | 0.559   | 0.281   | 0.282	| 0.480	| 0.645
> McSp+RL       | 0.812   | 0.559   | 0.269   | 0.432	| 0.476	| 0.633
> GLSearch      | 1.000   | 1.000   | 1.000   | 1.000	| 1.000	| 1.000
> --------------+---------+---------+---------+----------+----------+---------
> Best Sol Size | 128 	| 256 	| 509 	| 999  	| 2040 	| 4033
> ````````````~`~~
>
> Section E.3 in the latest submission addresses this.

---

### Author Response · Authors · 2020-11-17
**We have updated the PDF submission.**

We would like to sincerely thank all reviewers for your constructive feedback! We have improved our submission by updating our PDF submission. We have merged the supplementary mateiral with the main text into a single PDF.

Authors of Paper 370

---

### Author Response · Authors · 2020-11-24
**Paper Revision**

We thank all reviewers for their feedback! We have uploaded a revised version incorporating the suggested changes and new experiment results as mentioned in our official responses to the reviews.

One major improvement is the experimental verification of the real running time of our method, showing that the proposed method can find a larger common subgraph than all the baselines in fewer iterations AND less real running time.

---

### Decision · Program_Chairs · 2021-01-07
**Final Decision**

**Decision:**

Reject

**Comment:**

The paper present a new learning-based approach` to solve the Maximum Common Subgraph problem. All the reviewers find the idea of using GCN and RL to guide the branch and bound interesting although, even after reading the rebuttal, there are some important concerns about the paper.

The main issue raised by many reviewers are on scalability of the methods and motivation of the problem. It would be nice to add a scalability experiments on large networks(>1M nodes) to show that the method could potentially scale. In fact, the original motivation based on drug discovery, chemoinformatics etc. application is a bit weak because in those area domain specific heuristic should work better.

Overall, the paper is interesting but it does not meet the high publication bar of ICLR.